# Partial loss of CFIm25 causes learning deficits and aberrant neuronal alternative polyadenylation

Callison E Alcott[1,2,3], Hari Krishna Yalamanchili[2,4], Ping Ji[5], Meike E van der Heijden[2,6], Alexander Saltzman[7], Nathan Elrod[5], Ai Lin[5,8], Mei Leng[7], Bhoomi Bhatt[7], Shuang Hao[2,9], Qi Wang[2,9], Afaf Saliba[2,4], Jianrong Tang[2,9], Anna Malovannaya[7,10,11,12], Eric J Wagner[5], Zhandong Liu[2,9,13], Huda Y Zoghbi[1,2,4,6,14,15]*

[1]Program in Developmental Biology, Baylor College of Medicine, Houston, United States; [2]Jan and Dan Duncan Neurological Research Institute at Texas Children's Hospital, Houston, United States; [3]Medical Scientist Training Program, Baylor College of Medicine, Houston, United States; [4]Department of Molecular and Human Genetics, Baylor College of Medicine, Houston, United States; [5]Department of Biochemistry & Molecular Biology, University of Texas Medical Branch, Galveston, United States; [6]Department of Neuroscience, Baylor College of Medicine, Houston, United States; [7]Verna and Marrs McLean Department of Biochemistry and Molecular Biology Baylor College of Medicine, Houston, United States; [8]Department of Etiology and Carcinogenesis, National Cancer Center/Cancer Hospital, Chinese Academy of Medical Sciences and Peking Union Medical College, Beijing, China; [9]Section of Neurology, Department of Pediatrics, Baylor College of Medicine, Houston, United States; [10]Department of Molecular and Cellular Biology, Baylor College of Medicine, Houston, United States; [11]Mass Spectrometry Proteomics Core, Baylor College of Medicine, Houston, United States; [12]Dan L Duncan Comprehensive Cancer Center, Baylor College of Medicine, Houston, United States; [13]Graduate Program in Quantitative and Computational Biosciences, Baylor College of Medicine, Houston, United States; [14]Department of Pediatrics, Baylor College of Medicine, Houston, United States; [15]Howard Hughes Medical Institute, Baylor College of Medicine, Houston, United States

*For correspondence:
hzoghbi@bcm.edu

**Abstract** We previously showed that *NUDT21*-spanning copy-number variations (CNVs) are associated with intellectual disability (Gennarino et al., 2015). However, the patients' CNVs also included other genes. To determine if reduced *NUDT21* function alone can cause disease, we generated *Nudt21*[+/-] mice to mimic *NUDT21*-deletion patients. We found that although these mice have 50% reduced *Nudt21* mRNA, they only have 30% less of its cognate protein, CFIm25. Despite this partial protein-level compensation, the *Nudt21*[+/-] mice have learning deficits, cortical hyperexcitability, and misregulated alternative polyadenylation (APA) in their hippocampi. Further, to determine the mediators driving neural dysfunction in humans, we partially inhibited *NUDT21* in human stem cell-derived neurons to reduce CFIm25 by 30%. This induced APA and protein level misregulation in hundreds of genes, a number of which cause intellectual disability when mutated. Altogether, these results show that disruption of *NUDT21*-regulated APA events in the brain can cause intellectual disability.

## Introduction

The brain is acutely sensitive to the dose of numerous proteins, such that even small perturbations in their levels can cause neurological disease. Proteins that affect the expression of other genes, such as transcriptional or translational regulators, are particularly critical (*De Rubeis et al., 2014*; *Vissers et al., 2016*; *Yin and Schaaf, 2017*). Canonical examples of such broad regulators are the RNA-binding protein FMR1, which underlies fragile X syndrome, and the chromatin modulator MeCP2, whose loss or gain respectively causes Rett syndrome or *MECP2*-duplication syndrome (*Amir et al., 1999*; *Pieretti et al., 1991*; *Verkerk et al., 1991*).

Alternative polyadenylation (APA) is an important mechanism of transcript and protein-level regulation. Nearly 70% of human genes have multiple polyadenylation [poly(A)] sites, usually within the 3′ untranslated region (UTR) of the mRNA, allowing for transcript isoforms with different 3′ UTR lengths and content (*Derti et al., 2012*; *Hoque et al., 2013*). The general implication is that longer 3′ UTRs contain additional regulatory motifs for miRNA and RNA-binding protein binding sites. This allows for differential regulation of those gene products through mechanisms such as altered translation efficiency, mRNA stability, and transcript localization (reviewed in *Tian and Manley, 2017*). Alternative poly(A) site usage has been observed in multiple biological contexts, both in different disease states and in normal physiology and development (*Hoque et al., 2013*; *Masamha and Wagner, 2018*). In general, as cells differentiate, pathway-specific mRNAs increasingly use distal polyadenylation sites, while cells that are more proliferative use more proximal poly(A) sites (*Hoque et al., 2013*; *Ji et al., 2009*; *Sandberg et al., 2008*). In the brain, average 3′ UTR length increases throughout neurogenesis, and mature, post-mitotic neurons have the longest average 3′ UTR length of any cell type (*Guvenek and Tian, 2018*; *Miura et al., 2013*; *Zhang et al., 2005*). Moreover, neuronal mRNA 3′ UTR lengths are modulated by activity and vary between distinct cellular compartments (*Tushev et al., 2018*), suggesting that cleavage site selection in neurons needs to be carefully regulated and APA dysfunction could cause neurological symptoms.

Numerous genes regulate APA, but *NUDT21* is among the most consequential (*Gruber et al., 2012*; *Masamha et al., 2014*; *Tian and Manley, 2017*). *NUDT21* encodes CFIm25, a component of the mammalian cleavage factor I (CFIm) complex (*Kim et al., 2010*; *Rüegsegger et al., 1996*; *Yang et al., 2011*). CFIm25 binds UGUA sequences in pre-mRNA and the CFIm complex helps recruit the enzymes required for cleavage and polyadenylation (*Brown and Gilmartin, 2003*; *Rüegsegger et al., 1998*; *Yang et al., 2011*; *Yang et al., 2010*; *Zhu et al., 2018*). The UGUA binding sites are often enriched at the distal polyadenylation sites of *NUDT21*-regulated RNAs, so CFIm25 typically promotes the synthesis of longer mRNA isoforms (*Zhu et al., 2018*). Thus, when *NUDT21* expression is reduced, proximal cleavage sites are more frequently used. CFIm25 downregulation in multiple human and mouse cell lines typically causes 3′ UTR shortening in hundreds of genes, and a consequent increase in protein levels of a subset of those genes; however, there are numerous exceptions to these trends (*Brumbaugh et al., 2018*; *Gennarino et al., 2015*; *Gruber et al., 2012*; *Kubo et al., 2006*; *Li et al., 2015*; *Martin et al., 2012*; *Masamha et al., 2014*). Notably, *MECP2* is among the most affected genes in these cell-line studies, and slight perturbations in MeCP2 levels cause neurological disease (*Chao and Zoghbi, 2012*). Moreover, *NUDT21* is a highly constrained gene. In the Genome Aggregation Database (gnomAD) of ~140,000 putatively healthy individuals, 125 missense and 13 loss of function variants would be expected in *NUDT21* if loss of function were not pathogenic. Instead, there are only 15 missense and zero loss-of-function variants, suggesting that *NUDT21* loss of function is incompatible with health (*Lek et al., 2016*). Given this evidence, we hypothesized that *NUDT21* variants can cause neurological disease through APA misregulation of *MECP2* and other dose-sensitive genes in neurons.

Combining results from our previous work with data from the Decipher database, we have identified nine individuals with *NUDT21*-spanning duplications that have intellectual disability, and two patients with deletions that have both intellectual disability and seizures (*Firth et al., 2009*; *Gennarino et al., 2015*). However, the duplication patients also have three other genes common to their copy-number variations (CNVs) and the deletion patients have nearly 20 common genes (*Gennarino et al., 2015*). CNVs of these other genes could be causing their symptoms. Therefore, it is important to determine if *NUDT21* loss of function alone is sufficient to cause disease. Identifying

the disease-causing genes within CNVs facilitates more accurate diagnosis and prognosis, and allows for targeted therapy development. To that end, we generated *Nudt21*$^{+/-}$ mice to model the reduced CFIm25 expression observed in humans and assessed them for phenotypes similar to the symptoms seen in the deletion patients. We found that *Nudt21* heterozygosity causes a 50% loss of wild-type *Nudt21* mRNA as expected, but only a 30% reduction of its cognate protein, CFIm25. Consistent with what we observed in the deletion patients, we found that *Nudt21* heterozygosity is sufficient to cause learning deficits in mice in a variety of behavioral assays. Further, it causes cortical hyperexcitability and misregulated APA in the hippocampus. In addition, to see how *NUDT21* loss specifically affects neurons and how it might lead to human disease, we investigated *NUDT21*-depleted human neurons. We found that a 30% reduction of CFIm25 induces widespread abnormal APA and protein levels, including for a number of dose-sensitive, disease-associated genes. Altogether, these results provide important in vivo and human-specific evidence that reduced *NUDT21* expression can cause intellectual disability.

## Results

### *Nudt21* heterozygotes have 50% less *Nudt21* mRNA in their brain, but only 30% reduced CFIm25 protein

*Nudt21* expression has not been well explored within organisms, particularly in the brain. Thus, we first sought to confirm that mice express *Nudt21* in the brain, and found CFIm25 in neuronal (NeuN positive) nuclei in regions important for learning and memory—the cortex and hippocampus (*Figure 1A & B*). Notably, no non-neuronal brain cells visualized expressed CFIm25 at high enough levels to be detected by immunostaining (*Figure 1A & B*). Next, we generated *Nudt21* knockout mice by excising exons two and three, which removes the RNA-binding domain of CFIm25 and induces a frame shift mutation leading to nonsense-mediated decay of the transcript (*Figure 1C*; *Yang et al., 2011*). We crossed the *Nudt21*$^{+/-}$ mice with each other and observed that of the 32 pups born, there were no homozygous null offspring, showing that complete loss of CFIm25 is embryonic lethal (*Figure 1D*). Unexpectedly, while the heterozygous mice have the anticipated 50% reduction in wildtype *Nudt21* mRNA in whole-brain extracts, there was only ~30% reduction in CFIm25 protein (*Figure 1E & F*). These results demonstrate the successful knockout of one *Nudt21* copy and reveal a partial post-transcriptional compensation of CFIm25 protein levels. Finally, we found that the *Nudt21*$^{+/-}$ mice had a normal life span, but consistently weighed ~10% less than their wild-type littermates, indicating that *Nudt21* heterozygosity might cause some abnormalities (*Figure 1—figure supplement 1*).

### Partial loss of CFIm25 causes learning deficits

To determine if *Nudt21* heterozygosity affects cognitive function, we compared *Nud21*$^{+/-}$ mice and their wild-type littermates in several neurobehavioral assays starting at 30 weeks of age. We focused on learning and memory assays because intellectual disability is the most pronounced and consistent symptom seen in patients with *NUDT21*-spanning CNVs (*Gennarino et al., 2015*). We started with the conditioned fear test, which assesses the mice's ability to learn to associate an aversive event with a sensory context. In this test, we initially train the mice by exposing them to two tone-shock pairings in a novel chamber. Healthy mice will freeze as a behavioral expression of fear when they hear the tone for the second time. The following day, we return the mice to the same chamber, where healthy mice who learned to associate the chamber with the shock will freeze in recognition of the chamber. Lastly, we place the mice into a new chamber and replay the tone, where the healthy mice will remember that the tone is associated with the shock and freeze in fear. We found that *Nud21*$^{+/-}$ mice showed abnormal behavior throughout the assay. Their initial fear learning was attenuated: when exposed to the reinforcement sound cue played again two minutes after the initial tone-shock pairing during training, they froze on average ~50% less often than their wild-type littermates (*Figure 2Ai*). Furthermore, the following day, they froze on average ~30% less in both tests, indicating that *Nud21*$^{+/-}$ mice have reduced memory of the chamber and the sound cue compared to their wild-type littermates (*Figure 2Aii and iii*).

It is important to ensure that potential hyperactivity in the mutant mice does not confound the conditioned fear test results. If the mice are hyperactive, they may freeze less in the conditioned fear

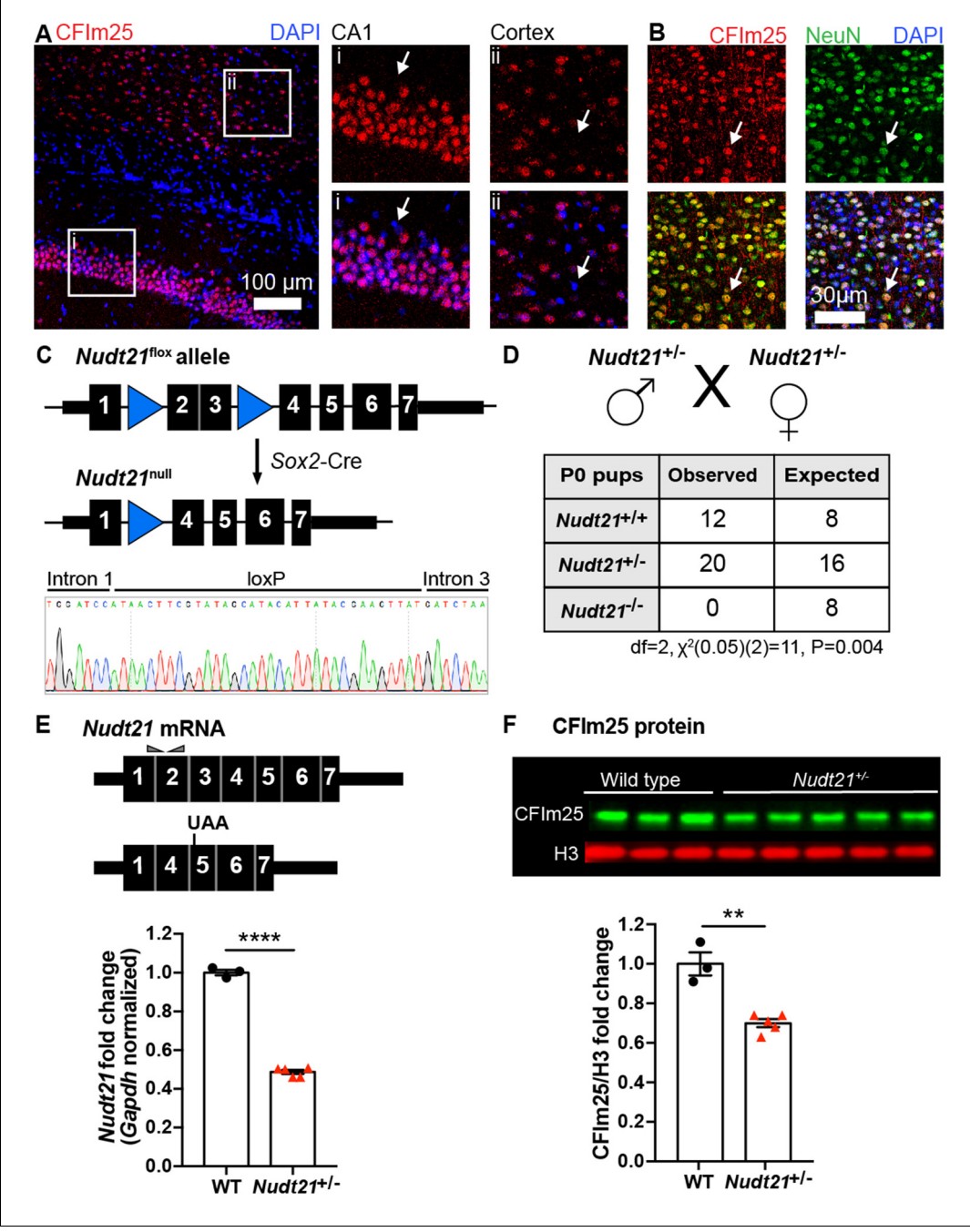

**Figure 1.** *Nudt21* heterozygotes have 50% less *Nudt21* mRNA in their neurons, but only 30% reduced CFIm25 protein. (**A**) Immunofluorescence showing CFIm25 expression in the nuclei of some cells in the (i) mouse hippocampus CA1 region and (ii) cortex. Arrows indicate nuclei that do not have CFIm25. (**B**) Immunofluorescence of the mouse cortex showing colocalization of CFIm25 and the neuronal marker NeuN. Arrows indicate a nucleus with CFIm25 and NeuN expression. (**C**) Schematic of the floxed and recombined *Nudt21* alleles (top), and sequencing showing successful recombination (bottom). (**D**) Observed and expected offspring counts born with each possible genotype from *Nudt21*[+/-] mating pairs. No homozygous *Nudt21* null offspring were born, when eight would be expected if loss of *Nudt21* did not affect survival: p=0.004, analyzed by two-tailed, chi-square test. (**E**) Schematic of wild-type and recombined *Nudt21* mRNA (top). Triangles indicate RT-qPCR primer binding sites, and UAA shows site of induced premature stop codon after recombination. RT-qPCR analysis shows expected 50% reduction of whole-brain, *Gapdh*-normalized, wild-type *Nudt21* mRNA in five-week-old mice with one wild-type *Nudt21* allele and one recombined, null allele (bottom): p<0.0001, n = 3–5/genotype. (**F**) Western blot image

*Figure 1 continued on next page*

*Figure 1 continued*

comparing five-week-old *Nudt21*$^{+/-}$ mice CFIm25 protein levels with their WT littermates (top). Western blot analysis showing ~30% reduction of H3-normalized CFIm25 protein levels in *Nudt21*$^{+/-}$ mice: p=0.0012, n = 3–5/ genotype. We confirmed that CFIm25 does not regulate H3. For all charts, error bars indicate SEM. All data analyzed by unpaired, two-tailed t-test unless otherwise stated. **p<0.01; ****p<0.0001. Weights of the heterozygous animals are shown in *Figure 1—figure supplement 1*.
The online version of this article includes the following figure supplement(s) for figure 1:

**Figure supplement 1.** *Nudt21*$^{+/-}$mice weigh less.

assay even if they remember the chamber or the cue. Therefore, we tested our mice in the open field assay. We place the mice in an open chamber and record the distance they travel in 30 min. We also record how much time they spend in the exposed center part of the chamber compared to the more hidden periphery. In this assay, hyperactive mice cover a greater distance, and anxious mice spend more time at the perimeter of the chamber (*Crawley, 1985*). We found that the *Nud21*$^{+/-}$ mice were not hyperactive, thus validating the conditioned fear results by showing they are not confounded by hyperactivity, and showing that partial loss of *Nudt21* function does cause cognitive defects (*Figure 2Bi*). Intriguingly, the *Nud21*$^{+/-}$ mice spent ~15% more time in the exposed, center part of the open field, which suggests they have reduced anxiety and shows that they have neuro-psychiatric features beyond learning and memory deficits (*Figure 2Bii*).

To determine if the learning deficits we detected in the *Nud21*$^{+/-}$ mice extended to other learning paradigms, we tested spatial learning using the Morris water maze. In this assay, the mice are trained over four days to locate a hidden platform in a pool of water using visual cues provided on the walls of the room. We initially reveal the platform to the mice to confirm visual acuity. Then the mice are individually placed at different locations around the perimeter of the pool eight times per day and timed for how long it takes them to locate the platform. Mice with spatial learning deficits require more time to find the platform throughout training. For an additional measurement, the platform is removed after the training and the mice are placed in a novel location on the perimeter of the pool. We then record how much time, out of a minute, they spend searching for the platform in the quadrant of the pool where it had been. Mice with spatial learning and memory deficits spend relatively less time in the platform quadrant, indicating that they do not remember its location as well. We found that the *Nudt21*$^{+/-}$ mice indeed have spatial learning deficits. On average, they required more time to locate the platform during their training trials, despite swimming slightly faster and covering a greater distance (*Figure 2Ci and ii*). Moreover, when we removed the platform, the *Nudt21*$^{+/-}$ mice spent ~30% less time in the correct quadrant (*Figure 2Ciii and iv*). Thus, the conditioned fear and Morris water maze results show that *Nudt21*$^{+/-}$ mice have learning and memory deficits in multiple domains.

## *Nudt21*$^{+/-}$mice have increased cerebral spike activity

In addition to intellectual disability, the patients with *NUDT21*-spanning deletions had seizures. In general, though, mice are much less sensitive to seizure-causing mutations than humans, but seizure susceptibility can be assessed using electroencephalography (EEG) (*Amendola et al., 2014*; *Jiang et al., 1998*; *Kriscenski-Perry et al., 2002*; *Miura et al., 2002*). Despite a lack of detectable seizures in *Nudt21*$^{+/-}$ mice, we found significantly more EEG spikes in their frontal cortex relative to their wild-type littermates (*Figure 3*). This result indicates that *Nudt21* haploinsufficiency is alone sufficient to cause cortical hyperexcitability, and suggests that *NUDT21* loss of function might increase seizure risk.

## *Nudt21* heterozygotes have altered hippocampal alternative polyadenylation

The observations that individuals with *NUDT21*-spanning CNVs have intellectual disability (ID) and that *Nudt21*$^{+/-}$ mice have learning deficits provide strong evidence that *NUDT21* loss of function causes disease, but do not reveal the molecular pathology. Therefore, to understand how partial loss of *NUDT21* function might sicken neurons and cause ID, we assessed the consequences of

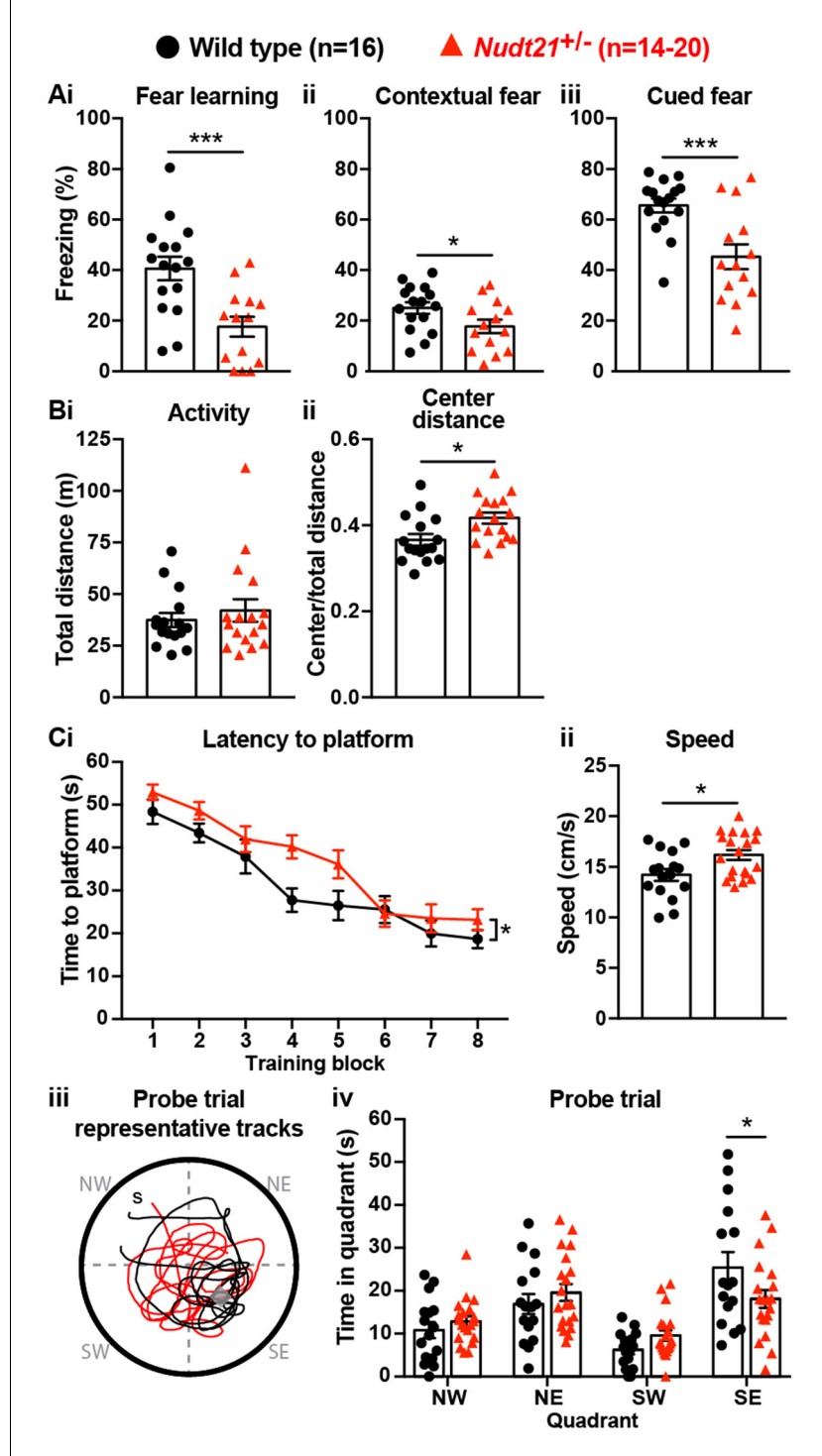

**Figure 2.** Partial loss of *Nudt21*-encoded CFIm25 protein causes learning deficits. (**A**) *Nudt21*[+/-] mice have conditioned fear learning deficits: fear learning, p=0.0009; contextual fear, p=0.045; cued fear, p=0.0009. (**B**) The open field assay shows that *Nudt21*[+/-] mice are no more active (i) but spend relatively more time in the center of the open field (ii), indicating reduced anxiety: p=0.01. (**C**) *Nudt21*[+/-] mice have spatial learning deficits in the Morris water maze. (i) They take longer to find the hidden platform during the training blocks (p=0.026, two-way, repeated measures ANOVA), (ii) despite swimming faster and farther (p=0.013). (iii and iv) When the hidden platform is removed in the probe trial, they spend less time in the quadrant that previously had the platform (p=0.039, Sidak's multiple comparisons test). For all assays, mice were between 30–40 weeks of age. Error bars

*Figure 2 continued on next page*

*Figure 2 continued*

indicate SEM. Representative tracks are from the animal with the median result in each genotype. All data were analyzed by unpaired, two-tailed t-test unless otherwise stated. *p<0.05; ***p<0.001.

partial *Nudt21* loss in the hippocampus for proteome and alternative polyadenylation (APA) dysregulation.

To assess the proteome, we used quantitative mass spectrometry, and to directly measure APA events, we used poly(A)-ClickSeq (PAC-seq), an mRNA 3′-end sequencing method that allows for more accurate identification of polyadenylation sites than standard RNA sequencing (*Figure 4—source data 1*; *Elrod et al., 2019*; *Routh et al., 2017*). Because we were studying heterozygotes and glia have low CFIm25 protein expression, we anticipated the changes would be small due to dilution effects from the glia. Nevertheless, we identified modest APA and proteomic changes in the *Nudt21*[+/-] mice, largely in neuronal genes (*Figure 4*). For example, the *Nudt21*[+/-] mice express more of the short isoform and less of the long isoform of *Ddx6*, a gene highly expressed in neurons and associated with autosomal dominant intellectual disability (*Figure 4Ai*; *Balak et al., 2019*; *Zhang et al., 2014*). We confirmed the altered mRNA isoform distribution by RT-qPCR (*Figure 4Aii–iii*). Overall, the genotypes separate by principle component analysis (PCA) when looking at global poly(A) site distribution, showing that the APA consequences of *Nudt21* heterozygosity in mice are reproducible (*Figure 4B*). Moreover, 20 genes have significantly different poly(A) site distributions despite the effects of neuronal CFIm25 reduction being diluted by mRNA from the glia (*Figure 4C*). Likewise, PCA shows a separation between the genotypes when looking at the whole proteome by quantitative mass spectrometry (*Figure 4D*). These proteomic changes reflect some of the direct effects of aberrant APA on translation efficiency and mRNA half-life, but most likely also include secondary effects, such as a reaction to dysfunctional neurons. Intriguingly, the PCA data from the mutant mice cluster more tightly than those of their wild-type littermates, perhaps because the effect on expression of partial *Nudt21* loss is strong enough to override natural variability. Lastly, of

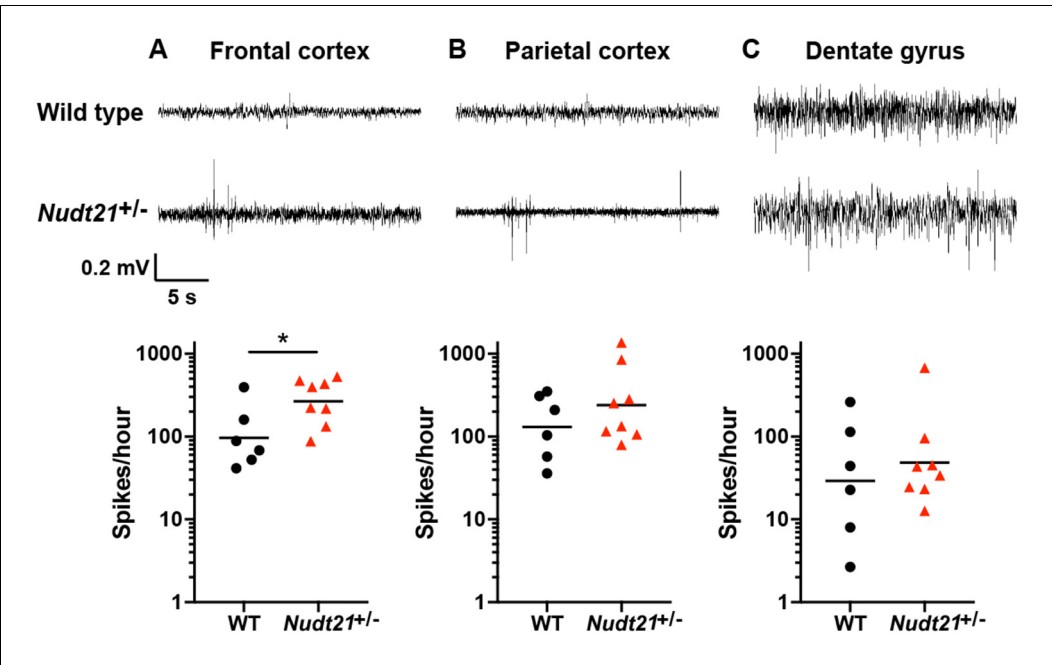

**Figure 3.** *Nudt21*[+/-] mice have increased cerebral spike activity. (**A**) 57-week-old *Nudt21*[+/-] mice have significantly increased spike activity in the frontal cortex by EEG (p=0.029), but not in the parietal cortex (**B**) and dentate gyrus (**C**). Representative traces are on top and spike count summaries below. All data analyzed by two-tailed Mann-Whitney test. Central tendency lines show the geometric mean. *p<0.05.

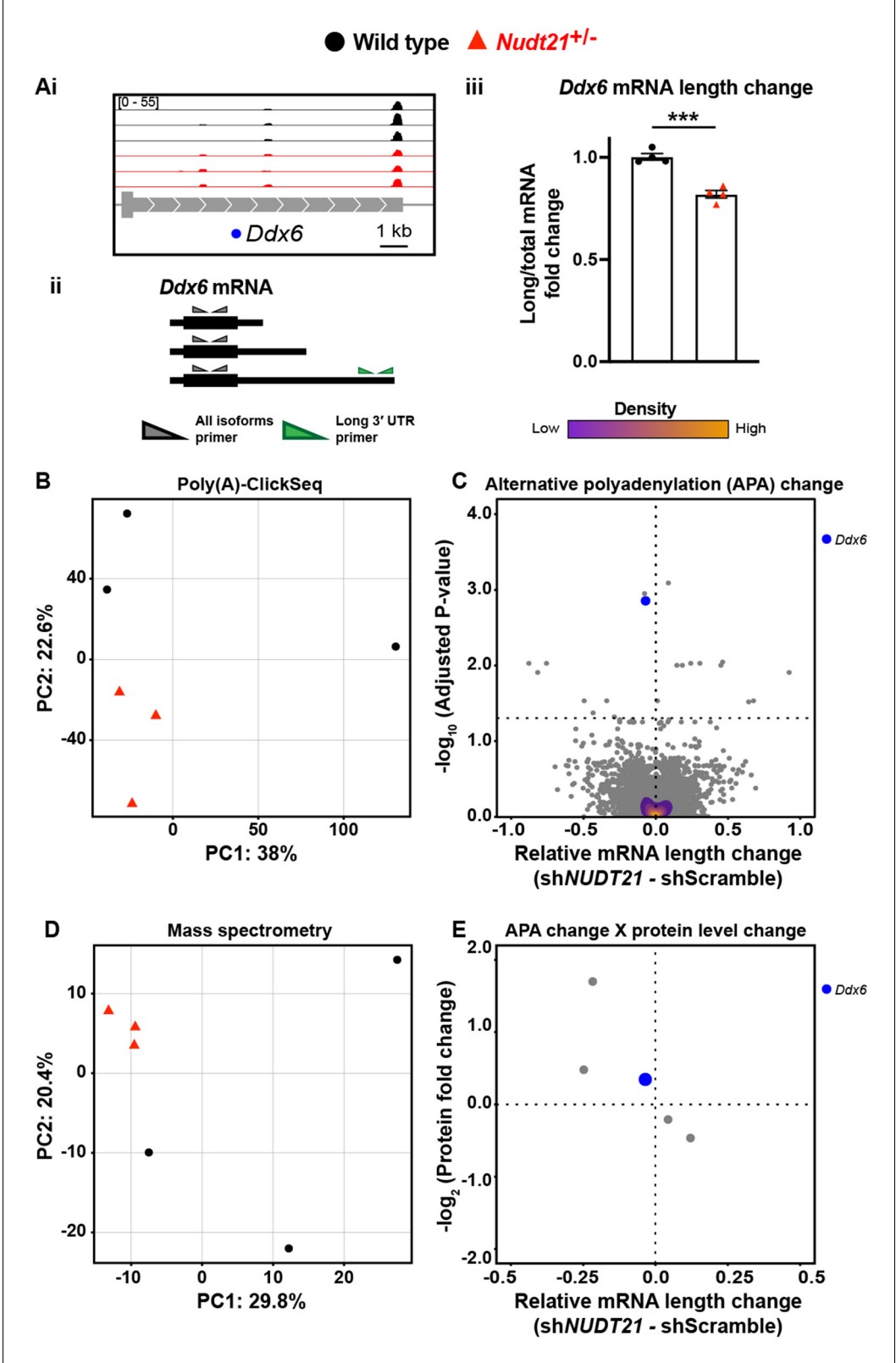

**Figure 4.** *Nudt21* heterozygotes have altered hippocampal alternative polyadenylation. (**A**)(i) Representative poly (A)-ClickSeq (PAC-seq) track showing altered alternative polyadenylation in *Ddx6*: n = 3/treatment. Peaks are 3′ end sequencing reads. Each peak indicates a different mRNA isoform with a different cleavage and polyadenylation site. Bracketed numbers show counts per million; kb stands for kilobases. (ii) Schemata of the
*Figure 4 continued on next page*

*Figure 4 continued*

three *Ddx6* mRNA isoforms identified by PAC-seq. Triangles show the binding sites of the qPCR primers used to validate the PAC-seq results. The gray primers detect all the *Ddx6* mRNA isoforms, whereas the green primers only detect the isoform with the longest 3′ UTR. (iii) *Gapdh*-normalized RT-qPCR quantification showing relatively less of the long isoform of *Ddx6* in the hippocampi of 46-week-old *Nudt21*$^{+/-}$ mice compared to their wild-type littermates. Error bars indicate SEM. Data analyzed by unpaired, two-tailed t-test. p=0.0003. (B) *Nudt21*$^{+/-}$ and wild-type mice separate by principle component analysis (PCA) of poly(A) site usage. (C) Volcano plot showing relative mRNA length change in *Nudt21*$^{+/-}$ mice compared to their wild-type littermates. The horizontal, dashed line shows $P_{adjusted}$ = 0.05, n = 3/genotype. (D) *Nudt21*$^{+/-}$ and wild-type mice segregate by PCA of the proteome by quantitative mass spectrometry. (E) Mass spectrometry quantification of protein level fold change for genes with significantly altered APA ($P_{adjusted}$ <0.05). Source files for the PAC-seq and mass spectrometry quantification data are available in *Figure 4—source data 1*. ***p<0.001.

The online version of this article includes the following source data for figure 4:

**Source data 1.** RNA length and protein level changes in genes with misregulated APA in *Nudt21*$^{+/-}$mice.

the 20 genes with measurable APA changes in whole-hippocampal lysate, five were detected by mass spectrometry and had protein level changes, including Ddx6 (*Figure 4E*). Although there are too few data points to perform a statistical analysis, all observations follow the trend seen in cell lines, where protein levels increase with mRNA shortening.

## *NUDT21* depletion induces aberrant alternative polyadenylation and altered protein levels in human neurons

Although the hippocampi of *Nudt21*$^{+/-}$ mice showed some molecular changes, we were most interested in evaluating the dysfunction in human neurons with partial *NUDT21* loss to better model cells from patients with *NUDT21* deletions or loss-of-function variants. Therefore, we generated human embryonic stem cell (ESC)-derived glutamatergic neurons and transduced them with either control shRNA or shRNA targeting *NUDT21* . Like the whole-brain extracts from *Nudt21*$^{+/-}$ mice, the human neurons had a 30% reduction of CFIm25 protein, despite a greater reduction of *NUDT21* mRNA (*Figure 5A & B*). This difference shows that there is also homeostatic stabilization of CFIm25 in human neurons.

As with the mouse hippocampi, we performed PAC-seq and quantitative mass spectrometry on the shRNA-infected human neurons (*Figure 6—source data 1*). PAC-seq shows reduced *NUDT21*, confirming the RT-qPCR results (*Figure 6Ai*). Similar to data observed in the *NUDT21* CNV patient lymphoblasts, *NUDT21* loss of function also results in the 3′ UTR shortening of *MECP2* in human neurons, as well as other genes strongly regulated by *NUDT21* in cancer-cell-line assays, such as *VMA21* and *PAK1* (*Figure 6Aii-iv*; *Chu et al., 2019*; *Gennarino et al., 2015*; *Masamha et al., 2014*). We validated the PAC-seq results using RT-qPCR (*Figure 6Bi–iii*).

To strengthen the validity of our analysis, we assessed the location of our poly(A) site calls within the gene, and found that >90% are in the 3′ UTR region as expected (*Figure 6—figure supplement 1A*). We also performed a PCA on the poly(A) site counts, and the samples separated by genotype (*Figure 6—figure supplement 1B*). Like in the mice, the neurons with partial *NUDT21* loss cluster more tightly than the controls.

Additionally, we looked at the distribution of the CFIm25 binding motif, UGUA, surrounding our poly(A) site calls. CFIm25 binds UGUA on the mRNA to promote cleavage and polyadenylation, so genes affected by *NUDT21* loss should have more of the UGUA motif upstream of the poly(A) site that is used less frequently in the *NUDT21*-depleted cells. As predicted, we found an enrichment of the UGUA sequence upstream of the distal cleavage site in *NUDT21*-regulated mRNAs that shortened after *NUDT21* knockdown, almost no difference in UGUA frequency upstream of proximal and distal cleavage sites in mRNAs not regulated by *NUDT21*, and a slight UGUA enrichment upstream of the proximal cleavage site in those mRNAs that lengthened after *NUDT21* knockdown (*Figure 6—figure supplement 1C*).

Overall, *NUDT21* loss of function in neurons, similar to other cell lines, causes widespread dysregulation of alternative polyadenylation, predominantly resulting in mRNA shortening (*Figure 6C*; *Brumbaugh et al., 2018*; *Gruber et al., 2012*; *Masamha et al., 2014*). Of the misregulated genes, 129 have a gnomAD probability of loss-of-function intolerance over 0.9, 26 cause

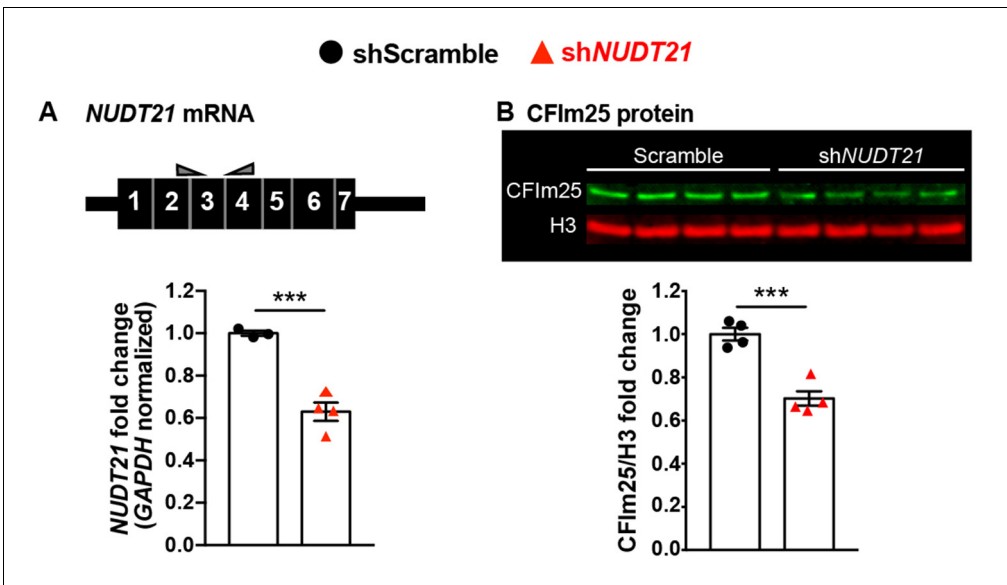

**Figure 5.** Human embryonic stem cell derived neurons infected with shRNA targeting *NUDT21* have 30% less CFIm25 protein. (**A**) Schematic of *NUDT21* mRNA with qPCR primers (top) and RT-qPCR quantification of *NUDT21* mRNA levels in human embryonic stem cell (ESC)-derived neurons infected with scrambled shRNA (shScramble) or shRNA targeting *NUDT21* (sh*NUDT21*). The neurons infected with sh*NUDT21* have a 40% reduction of *GAPDH*-normalized *NUDT21*: p=0.0009, n = 3–4/treatment. (**B**) Western blot image and quantification showing a 30% reduction of H3-normalized CFIm25 protein in sh*NUDT21*-infected neurons: p=0.0005, n = 4/treatment. We confirmed that CFIm25 does not regulate H3. ***p<0.001.

intellectual disability when mutated, and of those, nine are autosomal or X-linked dominant (*Supplementary file 1*; *Lek et al., 2016*; *McKusick-Nathans Institute of Genetic Medicine and Johns Hopkins University, 2019*; *Vissers et al., 2016*). In addition to *MECP2*, misregulation of some of these genes is likely contributing to pathogenesis.

Notwithstanding disrupted RNA localization, the most apparent consequence of misregulated APA is altered protein levels (*Tian and Manley, 2017*). To see what effect altered APA had on protein levels, we performed quantitative mass spectrometry on the neurons. For many of the genes with significantly misregulated APA, we also see concordant changes in protein levels (*Figure 6D*). Most commonly, *NUDT21* loss results in shorter mRNAs and increased protein, such as with MeCP2, VMA21, and PAK1, which respectively had protein-level increases of 40%, 90%, and 100% (*Figure 6D*).

## Partial loss of *NUDT21* function in neurons leads to downstream transcriptomic and proteomic dysregulation

We expected that these protein level changes, along with the ensuing neuronal dysfunction, would have downstream consequences. To assess these potential effects, we performed a differential expression analysis on the PAC-seq data (*Figure 7—source data 1*) and found that the genotypes also separate by genotype with PCA (*Figure 7A*). Interestingly, altered APA had almost no direct effect on mRNA levels: only two genes have significant changes in both APA and expression. However, there are numerous differentially expressed genes (DEGs), most likely resulting from secondary effects following APA misregulation (*Figure 7B*). 21 of the DEGs have a pLI >0.9, and four are currently associated with ID (*Supplementary file 1*; *Lek et al., 2016*; *McKusick-Nathans Institute of Genetic Medicine and Johns Hopkins University, 2019*; *Vissers et al., 2016*).

These transcriptional alterations in turn lead to a corresponding change in protein levels, demonstrating the cascading effects of *NUDT21* loss on mRNA and protein levels (*Figure 7C*). When analyzing all proteins, PCA shows a segregation by genotype, revealing that reduced *NUDT21* has a notable downstream effect on the entire neuronal proteome (*Figure 7D*).

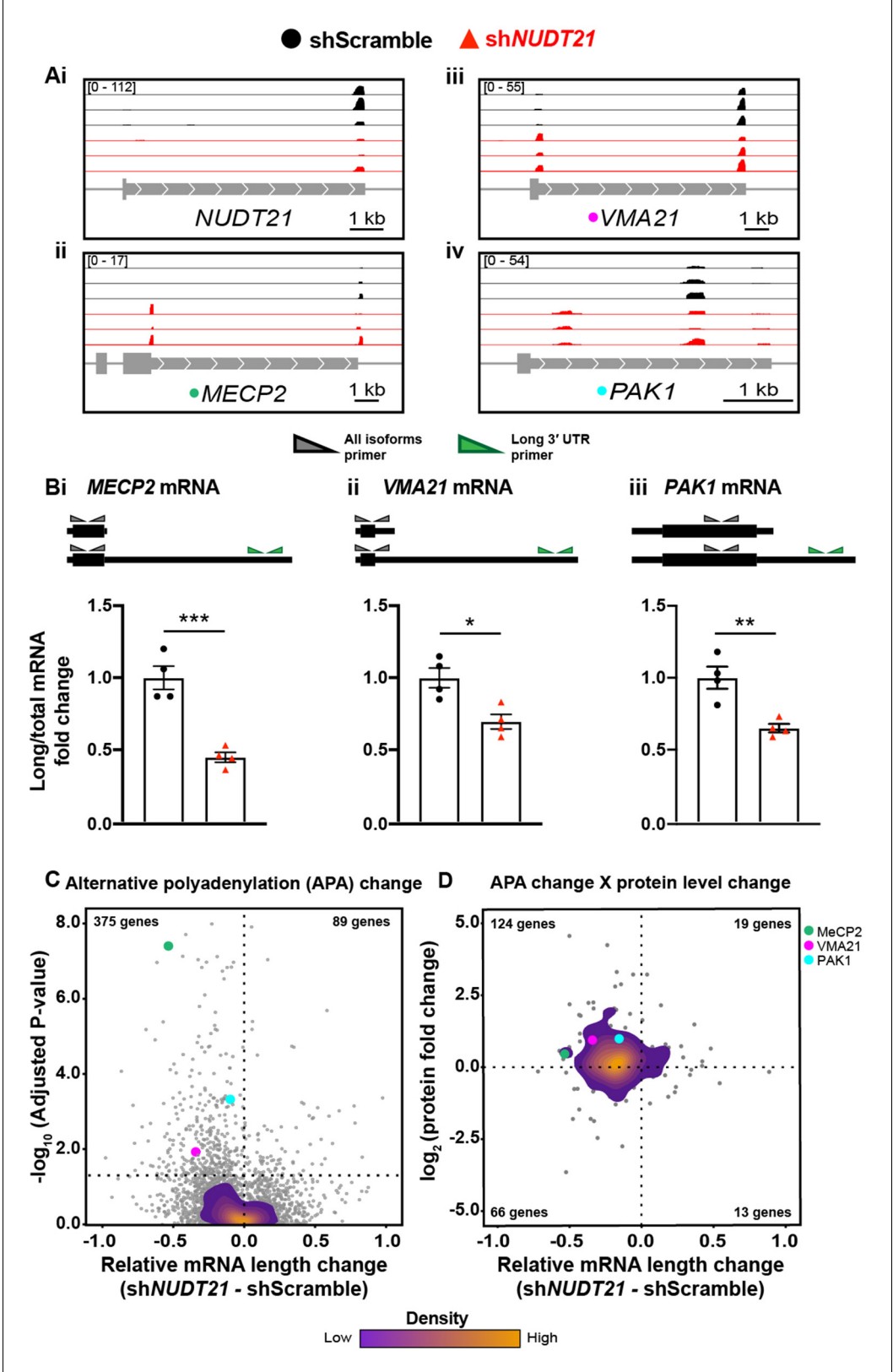

**Figure 6.** *NUDT21* depletion induces aberrant alternative polyadenylation and altered protein levels. (**A**)(i) Poly(A)-ClickSeq (PAC-seq) track showing reduced *NUDT21* and (ii-iv) PAC-seq tracks showing altered alternative polyadenylation in example genes *MECP2*, *VMA21*, and *PAK1*. n = 3/treatment. Peaks are 3′ end sequencing
*Figure 6 continued on next page*

*Figure 6 continued*

reads. Multiple peaks in a gene indicate multiple mRNA isoforms with different cleavage and polyadenylation sites. Bracketed numbers show counts per million; kb stands for kilobases. (**B**) Schemata of the mRNA isoforms identified by PAC-seq for *MECP2, VMA21,* and *PAK1* (top). Triangles show the binding sites of the qPCR primers used to validate the PAC-seq results. The gray primers detect all mRNA isoforms for the target gene, whereas the green primers only detect the isoform with the longest 3′ UTR. *Gapdh*-normalized RT-qPCR quantification showing relatively less of the long isoforms in the sh*NUDT21*-infected neurons (bottom) for *MECP2,* p=0.0007 (i), *VMA21,* p=0.01 (ii), and *PAK1,* p=0.005 (iii). Error bars indicate SEM. Data analyzed by unpaired, two-tailed t-test. (**C**) Volcano plot showing relative mRNA length change in sh*NUDT21*-infected neurons compared to shScramble-infected controls. The horizontal, dashed line shows $P_{adjusted}$ = 0.05, n = 3/treatment. *NUDT21* loss in neurons predominantly results in shorter mRNAs (p<0.0001, two-tailed chi-square test). >90% of reads are in 3′ UTRs (*Figure 6—figure supplement 1A*). Principle component analysis (PCA) shows sample separation by treatment (*Figure 6—figure supplement 1B*). Distal cleavage sites of *NUDT21*-regulated mRNAs are enriched for the CFIm25 binding motif, UGUA, in mRNAs that shorten after *NUDT21* knockdown, but not in non-target mRNAs (*Figure 6—figure supplement 1C*). (**E**) Mass spectrometry quantification of protein level fold change for genes with significantly altered APA ($P_{adjusted}$ <0.05). mRNA shortening predominantly results in increased protein levels (p<0.0001, two-tailed conditional chi-square test). Source files for the PAC-seq and mass spectrometry quantification data are available in *Figure 6—source data 1*. *p<0.05; **p<0.01; ***p<0.001.
The online version of this article includes the following source data and figure supplement(s) for figure 6:

**Source data 1.** RNA length and protein level changes in genes with misregulated APA following *NUDT21* inhibition in human neurons.
**Figure supplement 1.** Accessory analyses support PAC-seq validity.

While we did not previously know the consequences of *NUDT21* loss in post-mitotic neurons, we found that many of the *NUDT21*-regulated genes are the same as those seen in cancer-line studies, and the majority follow the same trends. Partial loss of *NUDT21* function in neurons broadly causes 3′ UTR shortening and increased protein levels that in turn leads to widespread transcriptomic and proteomic dysregulation, ultimately demonstrating the constitutive function of this gene in various cell types.

## Discussion

Here we show that a partial loss of *Nudt21* function causes learning deficits, cortical hyperexcitability, and APA and proteomic misregulation in mice. Further, we show in human neurons that partial loss of *NUDT21* function broadly disrupts gene expression via widespread misregulated alternative polyadenylation and protein levels. Alongside our previous discovery that patients with *NUDT21*-spanning deletions have intellectual disability and gnomAD data that *NUDT21* is a highly constrained gene, our results provide strong evidence that partial loss of *NUDT21* function causes intellectual disability (*Gennarino et al., 2015*; *Lek et al., 2016*).

Because a relatively small reduction in CFIm25 protein was sufficient to cause deficits, these data suggest that individuals with missense variants in *NUDT21* that affect its function may also have intellectual disability. Moreover, duplications of *NUDT21*, which we previously showed are associated with ID, should lead to comparable dysregulation of alternative polyadenylation and protein levels, including reduced MeCP2, and thus cause disease (*Gennarino et al., 2015*).

Beyond providing insight into *NUDT21*-associated disease, these data provide useful perspectives on the broader field of pediatric neurodevelopmental disease research. They illustrate the importance of protein-level homeostasis. A mere 30% reduction of CFIm25 protein in the mouse brain was sufficient to cause learning deficits, and mice are neurologically less sensitive to genetic insult than humans (*Tan and Zoghbi, 2019*). Further, many genes associated with neurodevelopmental diseases encode proteins that regulate or affect transcription (*De Rubeis et al., 2014*; *Vissers et al., 2016*; *Yin and Schaaf, 2017*). *NUDT21* adds another dimension to this group of genes insofar as it broadly affects alternative polyadenylation and protein levels. As a group, these genes demonstrate how a partial loss or gain of function can result in large effects, and further show that neurons are particularly sensitive to protein-level disequilibrium.

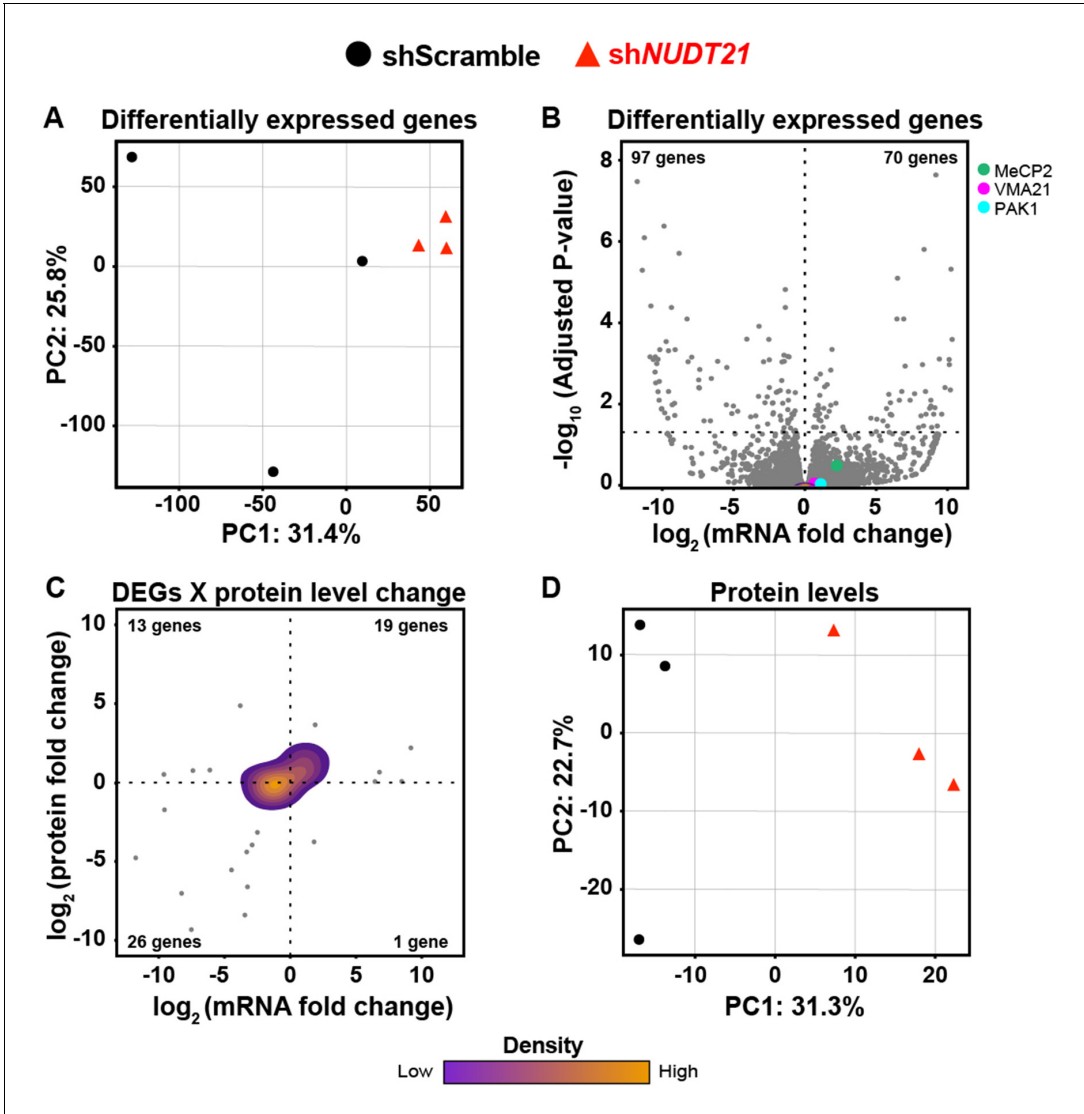

**Figure 7.** Partial loss of *NUDT21* function in neurons leads to downstream transcriptomic and proteomic dysregulation. (**A**) Control and sh*NUDT21*-infected human neurons separate by principle component analysis (PCA) of differentially expressed genes (DEGs). (**B**) Volcano plot showing DEGs in sh*NUDT21*-infected neurons compared to shScramble-infected controls. The horizontal, dashed line shows $P_{adjusted}$ = 0.05, n = 3/treatment. (**C**) Mass spectrometry quantification of protein level fold change for genes with significantly altered DEGs ($P_{adjusted}$ <0.05). (**D**) Control and sh*NUDT21*-infected human neurons segregate by PCA of the proteome. Source files for the differential gene expression and mass spectrometry quantification data are available in *Figure 7—source data 1*.

The online version of this article includes the following source data for figure 7:

**Source data 1.** Differential gene expression following *NUDT21* inhibition in human neurons.

Lastly, while the observed 40% increase of MeCP2 protein likely contributes to the symptoms seen in the *NUDT21* CNV patients, the dysregulation in neurons following *NUDT21* loss is so widespread that there are probably other mediators of the patients' condition. Most likely, patient symptoms result from the additive effects of numerous misregulated genes, especially those that are ID associated. This then points to normalization of CFIm25 levels as the most viable molecular therapeutic strategy for treating this disease.

# Materials and methods

## Key resources table

| Reagent type (species) or resource | Designation | Source or reference | Identifiers | Additional information |
|---|---|---|---|---|
| Gene (*Mus musculus*) | *Nudt21* | | NCBI Gene ID: 68219 | |
| Gene (*Homo sapiens*) | *NUDT21* | | NCBI Gene ID: 11051 | |
| Strain, strain background (*M. musculus*; male and female) | C57BL/6J | The Jackson Laboratory | IMSR Cat# JAX:000664, RRID:IMSR_JAX:000664 | |
| Genetic reagent (*M. musculus*) | *Sox2*-Cre | PMID:14595839 | IMSR Cat# JAX:008454, RRID:IMSR_JAX:008454 | |
| Genetic reagent (*M. musculus*) | *Nudt21*$^{flox/flox}$ | PMID:30830875 | RRID:MGI:6385511 | Dr. Eric Wagner (University of Texas Medical Branch) |
| Genetic reagent (*M. musculus*) | *Nudt21*$^{+/-}$ | This paper | RRID:MGI:6385902 | Line maintained in the H. Zogbhi lab |
| Cell line (*H. sapiens*) | HEK293T | ATCC | ECACC Cat# 12022001, RRID:CVCL_0063 | |
| Cell line (*H. sapiens*) | WA09 | WiCell | RRID:CVCL_9773 | |
| Antibody | Anti-CFIm25 (rabbit polyclonal) | Proteintech | Proteintech Cat# 10322–1-AP, RRID:AB_2251496 | 1:50 |
| Antibody | Anti-NeuN (mouse monoclonal) | Millipore | Millipore Cat# MAB377, RRID:AB_2298772 | 1:50 |
| Antibody | Anti-CFIm25 (mouse monoclonal) | Santa Cruz Biotechnology | Santa Cruz Biotechnology Cat# sc-81109, RRID:AB_2153989 | 1:500 |
| Antibody | Anti-H3 (rabbit monoclonal) | Cell Signaling Technology | Cell Signaling Technology Cat# 4499, RRID:AB_10544537 | 1:10,000 |
| Recombinant DNA reagent | pGIPZ | Open Biosystems | | shRNA backbone |
| Sequence-based reagent | *Nudt21* forward | This paper | qPCR primer | TACATCCAGCAGACCAAGCC |
| Sequence-based reagent | *Nudt21* reverse | This paper | qPCR primer | AATCTGGCTGCAACAGAGCT |
| Sequence-based reagent | *Gapdh* forward | This paper | qPCR primer | GGCATTGCTCTCAATGACAA |
| Sequence-based reagent | *Gapdh* reverse | This paper | qPCR primer | CCCTGTTGCTGTAGCCGTAT |
| Sequence-based reagent | *NUDT21* forward | This paper | qPCR primer | CTTCAAACTACCTGGTGGTG |
| Sequence-based reagent | *NUDT21* reverse | This paper | qPCR primer | AAACTCCATCCTGACGACC |
| Sequence-based reagent | *GAPDH* forward | This paper | qPCR primer | CGACCACTTTGTCAAGCTCA |
| Sequence-based reagent | *GAPDH* reverse | This paper | qPCR primer | TTACTCCTTGGAGGCCATGT |

*Continued on next page*

Continued

| Reagent type (species) or resource | Designation | Source or reference | Identifiers | Additional information |
|---|---|---|---|---|
| Sequence-based reagent | *Nudt21*null forward | This paper | PCR primer | ACAGATTAGCTGTTAGTACAGG |
| Sequence-based reagent | *Nudt21*null reverse | This paper | PCR primer | GAAGAACCAGAGGAAACGTGAG |
| Sequence-based reagent | Wild-type *Nudt21* forward | This paper | PCR primer | AGGAGGCTGACATGGATTGTT |
| Sequence-based reagent | Wild-type *Nudt21* reverse | This paper | PCR primer | TCTTCTCCTGGGTTAAGTTCCC |
| Sequence-based reagent | Anti-*NUDT21* shRNA 1 | Dharmacon | Clone ID: V2LHS_253272 | CAACCTGTACCCTCTTACCAATTATAC |
| Sequence-based reagent | Anti-*NUDT21* shRNA 2 | Dharmacon | Clone ID: V2LHS_197948 | ATCCATATATTCCTGCACATATT |
| Sequence-based reagent | Non-silencing shRNA | Dharmacon | Clone ID: RHS4348 | |
| Sequence-based reagent | P7 adapter (Illumina_4N_21T) | Illumina | Reverse transcription primer | GTGACTGGAGTTCAGACGTGTGCTCTTCCGATCTNNNNTTTTTTTTTTTTTTTTTTTTT |
| Sequence-based reagent | P5 adapter | Interated DNA Technologies | Reverse transcription primer | 5'HexynylNNNNAGATCGGAAGAGCGTCGTGTAGGGAAAGAGTGTAGATCTCGGTGGTCGCCGTATCATT |
| Sequence-based reagent | Universal primer | | cDNA amplification primer | AATGATACGGCGACCACCGAG |
| Sequence-based reagent | Example 3' indexing primer | | cDNA amplification primer | CAAGCAGAAGACGGCATACGAGATCGTGATGTGACTGGAGTTCAGACGTGT |
| Sequence-based reagent | *Ddx6* forward | This paper | qPCR primer | TGGATCTCATCAAGAAAGGC |
| Sequence-based reagent | *Ddx6* reverse | This paper | qPCR primer | GTGACAACAATTTATCTGCCTC |
| Sequence-based reagent | *Ddx6*-long forward | This paper | qPCR primer | CACAGCTGACAGACTCCAACA |
| Sequence-based reagent | *Ddx6*-long reverse | This paper | qPCR primer | AGCTTACTAACCCAGGCCCA |
| Sequence-based reagent | *MECP2* forward | This paper | qPCR primer | GATCAATCCCCAGGGAAAAGC |
| Sequence-based reagent | *MECP2* reverse | This paper | qPCR primer | CCTCTCCCAGTTACCGTGAAG |
| Sequence-based reagent | *MECP2*-long forward | This paper | qPCR primer | GCCTGGAAACCTGTCTGAGG |
| Sequence-based reagent | *MECP2*-long reverse | This paper | qPCR primer | CTCCAGCTAAGTGTGTCCCG |
| Sequence-based reagent | *VMA21* forward | This paper | qPCR primer | TACATATTTGAAGGCGCCC |
| Sequence-based reagent | *VMA21* reverse | This paper | qPCR primer | CATACACAAAGAGGGCCAG |
| Sequence-based reagent | *VMA21*-long forward | This paper | qPCR primer | AGGGGGAGGATTTGGATGTG |
| Sequence-based reagent | *VMA21*-long reverse | This paper | qPCR primer | TAGCTAAAGAACTCAAGCCCCC |
| Sequence-based reagent | *PAK1* forward | This paper | qPCR primer | GAATTACTTGGACAGTTACCTCG |
| Sequence-based reagent | *PAK1* reverse | This paper | qPCR primer | ACATCTGTCAAGGAGCCTC |

*Continued on next page*

*Continued*

| Reagent type (species) or resource | Designation | Source or reference | Identifiers | Additional information |
|---|---|---|---|---|
| Sequence-based reagent | *PAK1*-long forward | This paper | qPCR primer | CCAGCATTGTGGCTTGTCAT |
| Sequence-based reagent | *PAK1*-long reverse | This paper | qPCR primer | TTGTGCTGCAGAGGCAGTAG |
| Commercial assay or kit | QuantiTect Reverse Transcription Kit | Qiagen | Cat# 205313 | |
| Chemical compound, drug | 5-Fluoro-2′-deoxyuridine | Sigma-Aldrich | Product Number: F 0503 | 1 µM |
| Software, algorithm | LI-COR Image Studio | LI-COR | Image Studio Lite, RRID:SCR_013715 | |
| Software, algorithm | FreezeFrame 3 | Actimetrics | RRID:SCR_014429 | Conditioned fear assay |
| Software, algorithm | Fusion | Accuscan Instruments | RRID:SCR_017972 | Open field assay |
| Software, algorithm | EthoVision XT | Noldus Information Technology | RRID:SCR_000441 | Morris water maze |
| Software, algorithm | Clampfit 10 | Molecular Devices | RRID:SCR_011323 | EEG |
| Software, algorithm | Proteome Discoverer | Thermo Fisher Scientific | RRID:SCR_014477 | Mass spectrometry |
| Software, algorithm | Prism | Graphpad | RRID:SCR_002798 | |

## Generation of *Nudt21*$^{+/-}$ mice

The Baylor College of Medicine Institutional Animal Care and Use Committee approved all mouse care and manipulation (IACUC, protocol AN-1013). We generated C57BL/6J *Nudt21*$^{flox/flox}$ mice by inserting loxP sites flanking exons 2 and 3 of *Nudt21* by homologous recombination in embryonic stem cells (Ozgene) (*Weng et al., 2019*). We then crossed male *Nudt21*$^{flox/flox}$ mice with female C57BL/6J *Sox2*-Cre hemizygous mice (B6.Cg-*Edil3*$^{Tg(Sox2-cre)1Amc}$/J) to obtain *Nudt21*$^{+/-}$ mice. All oocytes from *Sox2*-Cre hemizygous females have Cre and can induce cre/lox recombination, even those haploid oocytes that do not have the Cre transgene (*Hayashi et al., 2003*; *Hayashi et al., 2002*). We confirmed successful cre/lox recombination by Sanger sequencing. To eliminate potential confounding from recombination mosaicism, we first crossed the *Nudt21*$^{+/-}$ F1 generation with wild-type C57BL/6J mice (Jackson Lab) before establishing mating pairs with F2s to generate experimental cohorts.

## Immunofluorescence

We dissected then hemisected brains from three seven-week-old *Nudt21*$^{flox/flox}$ C57BL/6J mice and drop-fixed their brains in 4% PFA overnight with gentle rocking at 4°C. After fixation, we cryoprotected the brains by gentle rocking in 30% sucrose solution in PBS at 4°C until the tissue sank. We then froze the brains in OCT, cryosected 40 µm sections, and stored them at 4°C.

We performed immunolabeling as previously described (*van der Heijden and Zoghbi, 2018*). Briefly, we first blocked free-floating sections in 5% normal goat serum, 0.5% Triton-X in PBS for one hour at room temperature. We then incubated the sections in siRNA-validated CFIm25 primary antibody (1:50, Proteintech 10322–1-AP) in blocking buffer overnight at 4°C, followed by three washes and Alexa Fluor 488 (1:1000, Molecular Probes) secondary antibody incubation for two hours at room temperature or anti-NeuN primary antibody (1:50, Millipore MAB377) and Alexa Fluor 555 (1:1000). Lastly, we labeled the nuclei with DAPI (1:10,000) and mounted the sections with Vectashield. We used a Leica TCS SP5 confocal microscope for imaging and ImageJ Fiji for image processing.

## RT-qPCR
### RNA extraction

We dissected and hemisected the brains of five-week-old and dissected the hippocampi of 46-week-old *Nudt21*$^{+/-}$ mice and their wild-type littermates. We immediately flash froze the tissue in liquid

nitrogen and stored it temporarily at −80°C. We later homogenized the tissue in TRIzol Reagent (ThermoFisher Scientific) with the Polytron PT 10–35 GT (Kinematica). We isolated RNA from the hemisected brains by chloroform phase separation, precipitated it with 2-propanol, and washed it with 75% ethanol. We eluted the purified RNA in water.

We extracted RNA from shRNA-infected, human ESC-derived neurons in a 12-well plate: four wells non-silencing scrambled (RHS4348) and eight wells sh*NUDT21* (V2LHS_253272 and V2LHS_197948) (Dharmacon). We lysed the neurons in the tissue-culture plate with TRIzol Reagent (ThermoFisher Scientific) and immediately transferred the lysate to microfuge tubes for trituration. We then isolated the RNA by chloroform phase separation, precipitation with 2-propanol, washing with 75% ethanol, and eluting in water as with the mice.

### RT-qPCR

We synthesized first-strand cDNA with M-MLV reverse transcriptase (Life Technologies) or for the PAC-seq validation experiments we used the QuantiTect Reverse Transcription Kit (Qiagen) because we could not use exon-spanning qPCR primers for amplifying the 3′ UTR region of example genes. We performed qPCR with PowerUp SYBR Green Master Mix (ThermoFisher Scientific) using the CFX96 Real-Time PCR Detection System (Bio-Rad). We designed exon-spanning primers using the UCSC genome browser mm10 and hg38 assemblies (http://genome.ucsc.edu/) (*Kent et al., 2002*), and Primer3 (http://bioinfo.ut.ee/primer3/) (*Koressaar and Remm, 2007*; *Untergasser et al., 2012*). For the PAC-seq validation experiments, we used GetPrime for primers to quantify total mRNA levels and Primer-BLAST for primers to quantify long 3′ UTR isoforms (*David et al., 2017*; *Ye et al., 2012*). We performed all RT-qPCR reactions in triplicate and determined relative cDNA levels by *NUDT21* threshold cycle (Ct) normalized to *GAPDH* Ct using the delta Ct method: relative expression (RQ) = $2^{-(NUDT21\ average\ Ct\ -\ GAPDH\ average\ Ct)}$. For showing relative shortening of PAC-seq example genes, we used primers that detected all of the mRNA for the gene along with additional primers that could only detect the long mRNA isoform. We then normalized the long isoform fold change to the total mRNA for each genotype. We present the data from human neurons infected with shRNA clone V2LHS_253272 because we used it for the PAC-seq experiment. We analyzed the data by two-tailed, unpaired t-test, and present it as mean ± SEM. *, **, ***, and **** denote p<0.05, p<0.01, p<001, p<0.0001.

### Western blot
#### Protein extraction

For the mice, we dissected and hemisected the brains of five-week-old *Nudt21*$^{+/-}$ mice and their wild-type littermates. We immediately flash froze them in liquid nitrogen and stored them temporarily at −80°C. We later homogenized the half brains with the Polytron PT 10–35 GT (Kinematica) in lysis buffer: 2% SDS in 100 mM Tris-HCl, pH 8.5, with protease and phosphatase inhibitors (ThermoFisher Scientific) and universal nuclease (Pierce). We incubated the lysates on ice for ten minutes, then rotated them for 20 min at room temperature. We spun the samples at top speed for 20 min to remove membrane, then quantified the protein levels with the Pierce Protein BCA Assay kit (ThermoFisher Scientific). For the ESC-derived neurons infected with shRNA targeting *NUDT21*, our extraction protocol was similar, except we lysed the cells directly in their 12-well plate and rocked them for 20 min at room temperature.

#### Western blot

We diluted the protein to 1 µg/µL in reducing buffer (LDS and sample reducing agent (ThermoFisher Scientific)) and ran 10 µg/sample. We imaged the membranes with the LI-COR Odyssey and analyzed the data with LI-COR Image Studio (LI-COR Biosciences), comparing H3-normalized CFIm25 levels. We confirmed the specificity of the CFIm25 antibody by sh*NUDT21* knockdown in HEK293T cells. We present the data from human neurons infected with shRNA clone V2LHS_253272 because we used it for the PAC-seq experiment. We analyzed the data by two-tailed, unpaired t-test, and present it as mean ± SEM. *, **, ***, and **** denote p<0.05, p<0.01, p<001, p<0.0001.

#### Antibodies
CFIm25: NUDT21 (2203C3): sc81109 (Santa Cruz); 1:500.

H3: Histone H3 (D1H2) XP (Cell Signaling Technology); 1:10,000.

## Mouse husbandry and handling

The Baylor College of Medicine Institutional Animal Care and Use Committee approved all mouse care and manipulation (IACUC, protocol AN-1013). We housed the mice in an AAALAS-certified level three facility on a 14 hr light cycle with ad libitum access to standard chow and water. We weighed a cohort of mice weekly from 4 to 12 weeks and a different cohort at 30 weeks. We tested both male and females in all experiments and only present the weight data separately because we never otherwise detected a difference between them. We were blinded to the mice's genotype during all handling.

## Genotyping

We determined the mice's genotypes by PCR amplification of tail lysates.

Primers:
Nudt21[null]
Forward: 5'- ACAGATTAGCTGTTAGTACAGG - 3'
Reverse: 5'- GAAGAACCAGAGGAAACGTGAG - 3'
Wild-type Nudt21
Forward: 5'- AGGAGGCTGACATGGATTGTT - 3'
Reverse: 5'- TCTTCTCCTGGGTTAAGTTCCC - 3'

## Behavioral tests

Because neurodevelopmental disease loss-of-function mouse models typically have more pronounced phenotypes when they are older, we started our behavioral battery on 30-week-old mice. We performed the assays on two cohorts. We tested the mice during their light cycle, typically between 11 AM and 5 PM. Prior to each assay, we habituated the mice to the testing facility for 30–60 min. The investigators were blind to the mice's genotypes during all assays.

## Conditioned fear

We first habituated the mice for 30 min in an adjacent room on each day of the test. On day one, we conditioned the mice by placing them in the habitest operant cage (Coulbourn) for a training session. The training consisted of two minutes habituation, then a 30 s 85 dB tone followed by a foot shock of 1.0 mA for two seconds. After another two minutes, we played the 30 s 85 dB tone again (the training day cue). Throughout the experiment, except for the two seconds during the foot shock, the FreezeFrame3 system (Coulbourn/Actimetrics) recorded the mice's movement and freezing episodes. On day two, we performed the contextual and cued fear assays. For contextual fear, we returned the mice to the test chambers precisely as we had done during the training, and recorded their freezing in the chamber for five minutes. We waited two hours before beginning the cued fear assay. We first changed the holding cages and test chamber shape, color, texture, scent, and lighting to make the experience as unrecognizable as possible to the mice. We then placed them in the modified chamber, and after three minutes played the 30 s 85 dB tone, then recorded their freezing for the following three minutes. We analyzed the difference in freezing between the two groups after the second sound cue (training cue) on day one, throughout the contextual fear test, and after the sound cue in the cued fear test, each by unpaired, two-tailed t-test. We excluded data for all the mice from one contextual fear trial. We suspect there was a technical error in the collection of those data: the FreezeFrame3 system recorded them as freezing far more than the mice in any other trial (50–80%) and two of the three were significant outliers in the Grubbs test.

## Open field

We lit the room to 200 lux and set the ambient white noise to 60 dB during habituation and throughout the test. We placed each mouse in the open field, a 40 × 40×30 cm chamber equipped with photobeams, and recorded their activity for 30 min with Fusion software (Accuscan Instruments). We analyzed total distance and center tolerance (center distance/total distance) by unpaired, two-tailed t-test.

## Morris water maze

Our Morris water maze experiment took place in a 120 cm diameter pool of water. We hid a 10 cm X 10 cm platform 0.5–1 cm underwater in the Southeast quadrant. On each wall of the testing room, we taped brightly colored shapes that the mice can use for orientation. Our experiment spanned four days. Each day, we set the lighting to 60 lux and habituated the mice in the testing room in their home cages for 30 min, then ten minutes in holding cages. Prior to the first day's experiment, we introduced the mice to the invisible platform by placing them on it for ten seconds. We next pulled the mice into the water and let them swim for ten seconds to ensure they could swim, then placed them directly in front of the platform to confirm they could climb back on to it. We tested the mice in two training blocks per day for the four days. Each training block consisted of four trials. For each trial, we started the mice in the pool at a different place on the perimeter (North, South, East, or West); the quadrant order was the same for every mouse in the trial, but different for every trial. We removed the mice after they found the platform, or if they did not find that platform, we guided them to the platform to rest on it for ten seconds before removing them. We used an Etho-Vision XT automated video tracking system (Noldus Information Technology) to track the mice's location, speed, and latency to find the platform. After the second training block on the fourth day, we immediately performed the probe trial: we removed the platform and placed the mice in a new location, the Northwest point on the perimeter of the pool, and tracked them for 60 s. We analyzed their speed by unpaired, two-tailed t-test; their latency to find the platform by two-way, repeated measures ANOVA (genotype*block); and their time in each quadrant in the probe trial by two-way, repeated-measures ANOVA (genotype*quadrant).

## Animal behavior statistical analysis

From previous experience, we know a sample size of 14 animals is sufficient to detect meaningful phenotypic differences in a neurobehavioral battery (*Chao et al., 2010*; *Lu et al., 2017*; *Samaco et al., 2013*). We analyzed all the data with Prism 7 (Graphpad), following Graphpad's recommendations. We used unpaired, two-tailed t-tests for all simple comparisons, and two-way repeated measures ANOVAs for all two-factor comparisons. We present all data as mean ± SEM. *, **, ***, and **** denote $p<0.05$, $p<0.01$, $p<0.001$, $p<0.0001$.

## Video electroencephalography (EEG) and spike counting

### Surgery and data recordings

The Baylor College of Medicine Institutional Animal Care and Use Committee approved all research and animal care procedures. We tested eight $Nudt21^{+/-}$ mice and six wild-type littermate controls. Experimenters were blind to the mouse genotype. We secured 54-week-old mice on a stereotaxic frame (David Kopf) under 1–2% isoflurane anesthesia. Each mouse was prepared under aseptic condition for the following recordings: the cortical EEG recording electrodes of Channels 1 and 2 were made of Teflon-coated silver wires (bare diameter 127 μm, A-M systems) and implanted in the subdural space of the parietal cortex and frontal cortex, respectively, with reference at the midline over the cerebellum. The electrode of the third channel, made of Teflon-coated tungsten wire (bare diameter 50 μm, A-M systems) was stereotaxically aimed at the hippocampal dentate gyrus (1.9 mm posterior, 1.7 mm lateral, and 1.8 mm below the bregma) with reference in the ipsilateral corpus callosum (*Paxinos and Franklin, 2001*). In addition, Teflon-coated silver wires were used to record the electromyogram (EMG) in the neck muscles to monitor mouse activity. All of the electrode wires together with the attached miniature connector sockets were fixed on the skull by dental cement. After two weeks of post-surgical recovery, mice received three two-hour sessions of EEG/EMG recordings over a week. Signals were amplified (100x) and filtered (bandpass, 0.1 Hz - 1 kHz) with the 1700 Differential AC Amplifier (A-M Systems), then digitized at two kHz and stored on disk for off-line analysis (DigiData 1440A and pClamp10, Molecular Devices). The time-locked mouse behavior was recorded by the ANY-maze tracking system (Stoelting Co.).

### EEG data analysis

Abnormal synchronous discharges were manually identified when the sharp positive deflections exceeded twice the baseline and lasted 25–100 ms (*Roberson et al., 2011*). We counted the number of abnormal spikes over the recording period using Clampfit 10 software (Molecular Devices, LLC)

and averaged the spike numbers across sessions for each animal. Since the data follow a lognormal distribution, we statistically compared the genotypes with the Mann-Whitney test using Prism 7 (Graphpad). The measure of central tendency is the geometric mean and * indicates p<0.05.

## shRNA lentivirus production and titer assessment
### Virus production
We made several viruses that express shRNAs targeting *NUDT21*. We transfected 45 ug DNA into 80–90% confluent, low-passage HEK293T cells (ATCC CRL-3216; RRID:CVCL_0063) in 150 mm dishes at a 4:3:1 ratio of pGIPz, psPAX2, pMD2.G with TransIT-293 transfection reagent (Mirus, MIR 2706). The following day, we changed the media to 10 mL. At 48 and 72 hr, we collected and pooled their media, then centrifuged at 4000 x g for ten minutes and filtered the supernatant through a poly-ethersulfone filter (VWR, 28145–505) to remove cellular debris. We concentrated the virus 100-fold with Lenti-X concentrator (Clontech, 631231) following the manufacturer's recommendations before aliquoting and freezing at −80°C.

### Titer assessment
We measured the viral titer using Open Biosystems' pGIPZ method (Thermo Fisher Scientific). We plated $5 \times 10^4$ HEK293T cells in a 24-well plate. The following day, we made a serial dilution of the virus in a 96-well plate, and when the HEK293T cells reached ~50% confluency, we infected them with the diluted virus. We cultured the cells for two days, then counted the tGFP colonies with an Axiovert 25 microscope (Zeiss) and X-cite 120 lamp (ExFo) to determine the viral titer. Our viruses had $10^9$ transducing units/mL.

### shRNA validation
Compared to non-silencing scrambled control virus (RHS4348, Dharmacon), we confirmed *NUDT21* knockdown efficacy in HEK293T cells (ATCC CRL-3216; RRID:CVCL_0063), and selected the two most efficient shRNAs for our studies: V2LHS_197948 and V2LHS_253272 (Dharmacon).

## Human embryonic stem cell (hESC)-derived neuron culture
We used WA09 (H9; RRID:CVCL_9773, WiCell) female embryonic stem cells (ESCs) to generate human neurons as previously described (*Jiang et al., 2017*). We confirmed their identity by STR analysis and verified that they were free of mycoplasma. We differentiated neural progenitors into human neurons over three weeks, changing the media every 3 days. Afterwards, we passaged the neurons with trypsin. Three days after passaging, we infected the neurons with lentiviruses containing pGIPZ shRNA clones at a multiplicity of infection of 10. We verified the tropism and infectivity of the virus using the tGFP reporter signal. On day two after infection, we treated the cells with 1 μM 5-Fluoro-2′-deoxyuridine for one day to remove proliferating glia from the culture (*Hui et al., 2016*). At day 3, we treated the neurons with puromycin (0.75–1.25 g/ml) for 6 days to select for infected cells. We cultured the cells for 60 days after infection, changing the media three times per week. We then aspirated all the media and washed the cells with PBS before freezing them at −80°C for later RNA and protein extraction.

## Poly(A) click-seq
### RNA extraction
We extracted RNA from the hippocampi of 46-week-old *Nudt21*$^{+/-}$ mice and their wild-type littermates as well as shRNA-infected, human ESC-derived neurons in a 12-well plate: four wells non-silencing scrambled (RHS4348) and eight wells sh*NUDT21* (V2LHS_253272) (Dharmacon). We know from past experience that we need at least three samples for RNA- and PAC-seq analysis (*Elrod et al., 2019*; *De Maio et al., 2018*; *Routh et al., 2017*; *Tan et al., 2016*). We prepared four samples per genotype to allow for loss of one sample. We lysed the tissue with TRIzol Reagent (ThermoFisher Scientific) and immediately transferred the lysate to microfuge tubes for trituration. We then isolated the RNA by chloroform phase separation, precipitation with 2-propanol, washing with 75% ethanol, and eluting in water.

## Library preparation and sequencing

We prepared libraries as previously described (*Routh et al., 2017*). We reverse transcribed 1 ug of total RNA with the partial P7 adapter (Illumina_4N_21T) and dNTPs with the addition of spiked-in azido-nucleotides (AzVTPs) at 5:1. We click-ligated the p5 adapter (IDT) to the 5′ end of the cDNA with CuAAC. We then amplified the cDNA for 21 cycles with Universal primer and 3′ indexing primer and purified it on a 2% agarose gel by extracting amplicon from 200 to 300 base pairs. We pooled the libraries and sequenced single-end, 150 base-pair reads on a Nextseq 550 (Illumina).

> P7 adapter (Illumina_4N_21T):
> GTGACTGGAGTTCAGACGTGTGCTCTTCCGATCTNNNNTTTTTTTTTTTTTTTTTTTTTT
> P5 adapter (IDT):
> 5′HexynylNNNNAGATCGGAAGAGCGTCGTGTAGGGAAAGAGTGTAGATCTCGGTGGTCGCCG
> TATCATT
> Universal primer:
> AATGATACGGCGACCACCGAG
> Example 3′ indexing primer:
> CAAGCAGAAGACGGCATACGAGATCGTGATGTGACTGGAGTTCAGACGTGT

## PAC-seq data analysis

We sequenced each library to a depth of >45 million for each human sample and >35 million for each mouse sample with 150 base pair (bp), single-end reads. For each sample, we obtained raw reads in a zipped fastq format. We used fastp for initial quality control (*Chen et al., 2018*). We filtered adapter contamination (AGATCGGAAGAGC) using the `-a` option. We trimmed the first six nucleotides and reads shorter than 40 nucleotides (nt) using the `-f` and `-l` options. We removed the poly(A) tail nucleotides using `hts_PolyATTrim` (https://github.com/ibest/HTStream) with parameters `-M 1 -x 0`.

## Read alignment

We downloaded raw FASTA sequences and annotations of the Human genome build GRCh38 from the UCSC table browser tool (*Karolchik, 2004*), and aligned trimmed reads to the reference genome with Bowtie 2 version 2.2.6 with parameters: `-D 20 R 3 N 0 L 20 -i S,1,0.50` (very-sensitive-local) (*Langmead and Salzberg, 2012*). We indexed the reference genome using bowtie2-build default settings, and saved sample-wise alignments as Sequence Alignment Map (SAM) files. We then used SAMtools V0.1.19 '*view*', '*sort*' and '*index*' modules to convert SAM files to Binary Alignment Maps (BAM), coordinate sort, and index (*Li et al., 2009*).

## Alternative poly-adenylation (APA) analysis

We computed strand-specific coverage of features (mRNA cleavage sites) from BAM files using the bedtools v2.25.0 '*genomecov*' module (*Quinlan and Hall, 2010*). We pooled sample-wise feature files to get a comprehensive list of polyadenylation (p(A)) sites across all the samples. Because mRNA cleavage is imprecise, we merged cleavage sites within 15 nucleotides of one another into a single feature using the bedtools '*merge*' module (*Tian and Graber, 2012*). We then mapped these features to known, annotated human polyadenylation sites downloaded from PolyA_DB3 (*Wang et al., 2018*). Because the annotations were in the hg19 coordinate system, we converted them to the GRCh38 coordinate system with the liftOver program from the UCSC genome browser portal. For subsequent analysis, we retained features with at least 1 bp of overlap with an annotated p(A) sites.

We quantified resultant p(A) sites across all samples using featureCounts (*Liao et al., 2013*). To minimize potential bias from under-covered p(A) sites, we retained only features with at least five reads in two samples and one read in the remaining sample in either of the conditions. We also filtered out p(A) sites accounting for less than 10% of the reads mapped to a gene in both the control and *NUDT21* knockdown conditions. We considered genes with mean read counts <5 in either condition as not expressed and excluded them.

Because about half of genes that undergo APA have greater than two p(A) sites, we wanted to use a statistic that could account for changes in all of the p(A) sites (*Derti et al., 2012*). Thus, we identified changes in alternative polyadenylation site usage using the Dirichlet multinomial test in

the DRIMSeq package, and computed adjusted p-values as previously described (*Nowicka and Robinson, 2016*; *Zhao et al., 2013*). We quantified the magnitude and direction of change in polyadenylation site usage as an mRNA lengthening score. We summed the weighted read counts per million (cpm) from each p(A) site in a gene to get that gene's mRNA length score. We assigned the weights in decreasing order from the most distal p(A) site as below:

$$W_i = 1 - \left( \left( \frac{1}{NS-1} \right) (i-1) \right)$$

where $W_i$ is the weight of $i^{th}$ distal polyadenylation site of a gene and $NS$ is the total number of polyadenylation sites mapped to it. Thus, in a gene with 3 p(A) sites, the cpm at the most promoter-distal site would all be counted (weight of 1) in the length score, the cpm at the intermediate site would be weighted by 0.5, and the cpm at the proximal site would not be counted (weight of 0). We then averaged the scores across samples and took the difference of treated to controls for the relative mRNA length change. We used a two-tailed, chi-square test to confirm enrichment of shorter mRNAs after *NUDT21* loss (p<0.0001) (*Figure 6C*).

## CFIm25 binding motif frequency analysis

Because about half of genes have more than two APA sites, for our proximal and distal site comparison, we analyzed the two sites with the maximum loss and gain in reads. We extracted strand specific sequences of these sites by extending the feature (i.e. the end of the sequencing read) by 200 nucleotides in the 5′ and 3′ directions. We computed UGUA motif frequency using a sliding window of five nucleotides and smoothed the frequencies using a one-dimensional Gaussian filter with a standard deviation parameter of 20. For a control comparison, we considered non-significant genes from the APA analysis with a fold change less than or equal to 0.05 to be non-targets.

## Poly(A) site distribution analysis

We used RSeQC to compute the read distributions across the 5′ UTR, coding sequence (CDS), intronic, and 3′ UTR regions (*Wang et al., 2012*).

## Differentially expressed genes analysis

We computed gene levels counts as the sum total of reads mapped to all annotated poly(A) sites from PolyA_DB3 within a gene (*Wang et al., 2018*). We excluded genes with an average read count of less than two across the samples. We then normalized raw read counts and tested for differential expression using DESeq2 as previously described (*Love et al., 2014*; *Yalamanchili et al., 2017*).

## Sample quality

We performed principle component analyses (PCA) on mRNA level and cleavage site counts of reads and confirmed that the genotypes separated (*Figure 4B and D*, *Figure 6—figure supplement 1B*, and *Figure 7A and D*; *Yalamanchili et al., 2017*).

## Disease association determination

For the probability of loss of function intolerance (pLI) of all the genes, we used gnomAD (*Lek et al., 2016*). In addition, we cross-referenced the misregulated genes with published genetic studies to see which were known to cause intellectual disability when mutated, then looked to the Online Mendelian Inheritance in Man (OMIM) database to see if the pathological variants were dominant or recessive (*McKusick-Nathans Institute of Genetic Medicine and Johns Hopkins University, 2019*; *Vissers et al., 2016*).

## Mass spectrometry

We processed, measured, and analyzed the sample as previously described (*Saltzman et al., 2018*). We summarize the main steps below.

## Lysis and digestion

We pelleted and lysed the shRNA-infected human neurons with three freeze (LN2) and thaw (42°C) cycles in 50 µL of ammonium bicarbonate + 1 mM $CaCl_2$. We then boiled the lysate at 95°C for two minutes with vortexing at 20 s intervals. We digested 50 µg of total protein with a 1:20 solution of 1 µg/µL trypsin:protein overnight at 37°C with shaking and then again with a 1:100 solution of 1 µg/µL trypsin:protein for 4 hr. We extracted the peptides with 80% acetonitrile + 0.1% formic acid solution, followed by centrifugation at 10,000 g, and vacuum drying.

## Off-Line basic pH reverse phase peptide fractionation

A '15F5R' protocol was used for off-line fractionation as described before (*Saltzman et al., 2018*). We filled a 200 µl pipette tip with 6 mg of C18 matrix (Reprosil-Pur Basic C18, 3 µm, Dr. Maisch GmbH) on top of a C18 disk plug (EmporeTM C18, 3M). We dissolved 50 µg of vacuum-dried peptides with 150 µl of pH10 ABC buffer and loaded it onto the pre-equilibrated C18 tip. We eluted bound peptides with fifteen 2%-step 2–30% gradient of acetonitrile non-contiguously combined into five pools, and then vacuum dried.

## Mass spectrometry

We analyzed fractionated peptides on an Orbitrap Fusion mass spectrometer coupled with the Nanospray Flex ion sources and an UltiMate 3000 UPHLC (Thermo Fisher Scientific). For each run, we loaded approximately one microgram of peptide onto a two cm 100 µm ID pre-column and resolved it on a twelve cm 100 µm ID column, both packed with sub-two µm C18 beads (Reprosil-Pur Basic C18, Catalog #r119.b9.0003, Dr. Maisch GmbH). We maintained a constant flow rate for 100 min 2–28% B gradient elutions, where A is water and B is 90% acetonitrile, both with 0.1% formic acid.

## Proteome Discoverer (Mascot-based) Search and Protein Inference/ Quantification

We used the Proteome Discoverer software suite (PD version 2.0.0.802; Thermo Fisher Scientific) to search the raw files with the Mascot search engine (v2.5.1, Matrix Science), validate peptides with Percolator (v2.05), and provide MS1 quantification through PD's Area Detector Module. We matched MS1 precursors in a 350–10,000 mass range against the tryptic RefProtDB database digest (2015-06-10 download) with Mascot permitting up to two missed cleavage sites (without cleavage before P), a precursor mass tolerance of 20 ppm, and a fragment mass tolerance of 0.5 Da. We allowed the following dynamic modifications: Acetyl (Protein N-term), Oxidation (M), Carbamidomethyl (C), DeStreak (C), and Deamidated (NQ). For the Percolator module, we set the target strict and relaxed FDRs for peptide spectral matches (PSMs) at 0.01 and 0.05 (1% and 5%), respectively. We used gpGrouper (v1.0.040) for gene product inference and label-free iBAQ quantification with shared peptide distribution. We then median-normalized iBAQ values by sample for further analysis.

## Analysis

Because mass spectrometry fold change calculations are inaccurate at the threshold of detection, we removed all proteins from the analysis that did not have consistent expression in at least one genotype, that is all three samples had at least one detectable and quantified peptide. Further, since unquantified proteins artificially inflate fold change calculations, we imputed missing values with random draws from a Gaussian distribution downshifted two standard deviations from the mean of all the measured values with a constant scaling factor of one, based on the missing value imputation procedure performed in Perseus (*Tyanova et al., 2016*). We employed the moderated t-test to calculate p-values and $\log_2$ fold changes for differentially expressed proteins as implemented in the R package *limma*, and adjusted for multiple-hypothesis testing with the Benjamini–Hochberg procedure (*Benjamini and Hochberg, 1995*; *Ritchie et al., 2015*).

We then plotted $\log_2$ protein level fold change values against their relative mRNA length change or their differential gene expression. To show that *NUDT21* loss predominantly causes mRNA shortening and increased protein levels, we used a multi-step, conditional chi-square approach (*Figure 6C* and *Figure 6D*). We first performed a two-tailed, chi-square test showing enrichment of shorter mRNAs after *NUDT21* loss ($p < 0.0001$) (*Figure 6C*). Then we performed a two-tailed chi-

square test on mRNA length change and protein levels to confirm that the data is not evenly distributed (p<0.0001) (*Figure 6D*, quadrants I-IV). Lastly, we performed a conditional two-tailed chi-square test on the protein levels of shortened mRNAs, demonstrating that the shortened mRNAs predominantly have increased protein (p<0.0001) (*Figure 6D*, quadrants II and III). Together, these tests confirm that *NUDT21* loss results in an enrichment of shorter mRNAs with elevated protein levels.

We performed PCA on the proteome as previously described to confirm that the treatment groups separated (*Figure 4D* and *Figure 7D*; *Yalamanchili et al., 2017*).

## Data availability

The PAC-seq data are available in the NCBI Gene Expression Omnibus (GEO), accession number GSE142683. For the alternative polyadenylation analysis code, see *Supplementary file 2*. We have deposited the mass spectrometry proteomics data to the ProteomeXchange Consortium via the PRIDE partner repository with the dataset identifier PXD014842 (*Perez-Riverol et al., 2019*).

## Ethics

For animal experimentation, we housed up to five mice per cage on a 14-hour light cycle in a level 3, AALAS-certified facility and provided water and standard rodent chow ad libitum. The Institutional Animal Care and Use Committee for Baylor College of Medicine and Affiliates approved all procedures carried out in mice under protocol AN-1013.

## Acknowledgements

We would like to thank the Zoghbi and Wagner lab members for critical feedback. We would also like to thank the cores that provided services for the project: the Microscopy, Neuroconnectivity, Animal Behavior, and Human Neuronal Differentiation cores from the Jan and Dan Duncan Neurological Research Institute and BCM Intellectual and Developmental Disabilities Research Center (NIH U54 HD083092 from the Eunice Kennedy Shriver National Institute of Child Health and Human Development); the Next Generation Sequencing core at the University of Texas Medical Branch; and BCM Mass Spectrometry Core (supported by NIH P30 CA125123 and CPRIT RP170005). Lastly, we would like to thank our funders who made the work possible: the NIH National Institute of Neurological Disorders and Stroke—F30NS095449 (CEA), National Cancer Institute—R03-CA223893-01 (PJ), National Institute of General Medical Sciences—R01-GM134539 (EJW), the Howard Hughes Medical Institute (HYZ), and the Intellectual and Developmental Disabilities Research Center—NIH U54 HD083092 from the Eunice Kennedy Shriver National Institute of Child Health and Human Development (HYZ). The content is solely the responsibility of the authors and does not represent the official views of the Eunice Kennedy Shriver National Institute of Child Health and Human Development, the National Institutes of Health, or the Howard Hughes Medical Institute.

## Additional information

### Competing interests

Huda Y Zoghbi: Senior editor, *eLife*. The other authors declare that no competing interests exist.

### Funding

| Funder | Grant reference number | Author |
| --- | --- | --- |
| National Institute of Neurological Disorders and Stroke | F30NS095449 | Callison E Alcott |
| Intellectual and Developmental Disabilities Research Center | NIH U54 HD083092 | Huda Y Zoghbi |
| Howard Hughes Medical Institute | | Huda Y Zoghbi |

| Eunice Kennedy Shriver National Institute of Child Health and Human Development | | Huda Y Zoghbi |
|---|---|---|
| National Cancer Institute | R03-CA223893-01 | Ping Ji |
| National Institute of General Medical Sciences | R01-GM134539 | Eric J Wagner |

The funders had no role in study design, data collection and interpretation, or the decision to submit the work for publication.

## Author contributions

Callison E Alcott, Conceptualization, Data curation, Formal analysis, Funding acquisition, Validation, Investigation, Visualization, Methodology, Project administration; Hari Krishna Yalamanchili, Data curation, Software, Formal analysis, Validation, Visualization, PAC-seq and mass spectrometry analysis, interpretation, and presentation; Ping Ji, Validation, Investigation, Methodology, PAC-seq experiment; Meike E van der Heijden, Formal analysis, Validation, Investigation, Visualization, Methodology, Immunofluorescence; Alexander Saltzman, Bhoomi Bhatt, Formal analysis, Methodology, Mass spectrometry analysis; Nathan Elrod, Formal analysis; Ai Lin, Investigation; Mei Leng, Validation, Investigation, Mass spectrometry experiment; Shuang Hao, Qi Wang, Formal analysis, Validation, Methodology, EEG analysis; Afaf Saliba, Formal analysis, Validation, Investigation, Mouse behavior experiments; Jianrong Tang, Resources, Supervision, EEG supervisor; Anna Malovannaya, Resources, Formal analysis, Supervision, Methodology, Mass spectrometry supervisor; Eric J Wagner, Resources, Supervision, Methodology, PAC-seq supervisor; Zhandong Liu, Resources, Software, Supervision, Bioinformatics supervisor; Huda Y Zoghbi, Conceptualization, Resources, Supervision, Funding acquisition, Project administration, Project supervisor

## Author ORCIDs

Callison E Alcott (iD) https://orcid.org/0000-0003-4166-5634
Meike E van der Heijden (iD) http://orcid.org/0000-0003-0801-8806
Alexander Saltzman (iD) http://orcid.org/0000-0003-4166-9555
Nathan Elrod (iD) http://orcid.org/0000-0003-1310-6026
Huda Y Zoghbi (iD) https://orcid.org/0000-0002-0700-3349

## Ethics

Animal experimentation: For animal experimentation, we housed up to five mice per cage on a 14-hour light cycle in a level 3, AALAS-certified facility and provided water and standard rodent chow ad libitum. The Institutional Animal Care and Use Committee for Baylor College of Medicine and Affiliates approved all procedures carried out in mice under protocol AN-1013.

## Decision letter and Author response

Decision letter https://doi.org/10.7554/eLife.50895.sa1
Author response https://doi.org/10.7554/eLife.50895.sa2

# Additional files

## Supplementary files

• Supplementary file 1. Intellectual disability associations of genes with misregulated APA and differential gene expression following neuronal *NUDT21* inhibition. Alternative polyadenylation (APA), Differentially expressed gene (DEG), probability of loss of function intolerance (pLI), intellectual disability (ID), Online Mendelian Inheritance in Man (OMIM), autosomal recessive (AR), autosomal dominant (AD), X-linked dominant (XLD), X-linked recessive (XLR)

• Supplementary file 2. Alternative polyadenylation analysis code.

• Transparent reporting form

## Data availability

The PAC-seq data are available in the Gene Expression Omnibus, accession number GSE142683. For the alternative polyadenylation analysis code, see Supplementary file 2. We have deposited the mass spectrometry proteomics data to the ProteomeXchange Consortium via the PRIDE partner repository with the dataset identifier PXD014842 (Perez-Riverol et al., 2018).

The following datasets were generated:

| Author(s) | Year | Dataset title | Dataset URL | Database and Identifier |
|---|---|---|---|---|
| Alcott CE, Yala-manchili HK, Ji P, van der Heijden ME, Saltzman AB, Elrod N, Lin A, Leng M, Bhatt B, Hao S, Wang Q, Saliba A, Tang J, Malovan-naya A, Wagner EJ, Liu Z, Zoghbi HY | 2019 | Partial loss of CFIm25 causes aberrant alternative polyadenylation and learning deficits | https://www.ncbi.nlm.nih.gov/geo/query/acc.cgi?acc=GSE142683 | NCBI Gene Expression Omnibus, GSE135384 |
| Alcott CE, Yala-manchili HK, Ji P, van der Heijden ME, Saltzman AB, Elrod N, Lin A, Leng M, Bhatt B, Hao S, Wang Q, Saliba A, Tang J, Malovan-naya A, Wagner EJ, Liu Z, Zoghbi HY | 2019 | Partial loss of CFIm25 causes aberrant alternative polyadenylation and learning deficits | https://www.ebi.ac.uk/pride/archive/projects/PXD014842 | PRIDE, PXD014842 |

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
