## [Decision Letter]

Thank you for submitting your article "Partial loss of CFIm25 causes aberrant alternative polyadenylation and learning deficits" for consideration by *eLife*. Your article has been reviewed by Eve Marder as the Senior Editor, Sacha Nelson as the Reviewing Editor, and two reviewers. The following individuals involved in review of your submission have agreed to reveal their identity: Erin Schuman (Reviewer #2).

The reviewers have discussed the reviews with one another and the Reviewing Editor has drafted this decision to help you prepare a revised submission.

Summary:

The authors have followed up on a prior human genetics study implicating copy number variation in *Nudt21*, which encodes the polyadenylation factor CFIm25, in intellectual disability and seizure disorder. In this advance, they create a mouse model of the disorder and demonstrate that heterozygous loss of function results in learning deficits, EEG abnormalities, altered polyadenylation of many genes and altered abundance of associated proteins. The study is important both because it illuminates the pathophysiology of a human neuropsychiatric disorder and because it highlights the importance of dosage regulation (here only a 30% reduction in protein results from the heterozygous loss) especially when the affected gene is critical for regulating the abundance or function of many other gene products, in this case by regulating alternative polyadenylation.

Essential revisions:

The reviewers and editors were enthusiastic about the importance of the study, but several key concerns about the ability of the data to support the conclusions were raised. These centered mainly on the issue of sequencing and proteomics data quality and analysis. The concerns were mainly that some of the poly(A) ClickSeq reads may be erroneous and need to be removed. False positive peaks can lead to miscalculation of genes with non-stop decay. Both reviewers noted the large variation among control samples shown in Figure 4—figure supplement 1. The PCA plot showed variability among controls (PC2) that was nearly as large as the experimental effect. There were also concerns about the statistical validity of the assessment of protein abundance. One reviewer suggested repeating the RNA-seq for expression analysis. The other suggested performing northern blots to confirm identified 3'UTR isoforms and their differential regulation in the mutant animals. They also suggested the following quality controls and metrics:

- What algorithm was used to identify peaks?

- What measures were taken (filtering steps) to prevent internal priming (false poly(A) peaks)?

- How many reads/peaks per gene feature (5'UTR/CDS/3'UTR/introns) were detected?

- What is the number of peaks per known poly(A) signal (how many peaks do not have known poly(A) signals)?

- What is the peak distance from known poly(A) signals?

- What is the replica correlation between poly(A) peak counts?

- Are the detected peaks conserved in other species?

- What is the fraction of sequenced reads that fall into poly(A) peaks after filtering? (It would be an issue if more reads are filtered, than reads contributing to poly(A) peaks.)

*Reviewer #1:*

In this manuscript by Alcott et al., the authors report learning deficits in mice with reduced CFIm25 expression. The authors claim that the phenotype is attributable to mis-regulation of alternative polyadenylation and resultant protein expression changes. This work extends the previous work from the Zoghbi lab where they found copy number variations of genomic segments spanning CFIm25 led to intellectual disability and seizures in patients. Overall, the current work was well carried out, especially the phenotype part, and the results are important. However, there are several concerns the authors should address before the paper can be accepted for publication.

1) The authors claim that 15% of genes with altered APA may undergo non-stop decay. This is quite a substantial number. Can they rule out the possibility that this is due to technical artefact of the sequencing method? Can they verify these non-stop decay-causing poly(A) sites, either by 3'RACE (for some) or comparing their data to those from other sequencing methods?

2) They also observed a non-stop decay isoform for *NUDT21*. However, this does not appear congruent with the fact that there is 30% reduction of protein level but 50% mRNA reduction. Non-stop decay should reduce both mRNA and protein expression levels. Therefore, the mRNA reduction should be more than 50%. Also, how can they explain the increase protein production per mRNA?

*Reviewer #2:*

In this manuscript the authors examine in mice heterozygous for *NUDT21* to mimic the loss of the human protein associated with various copy number variations. *NUDT21* mRNA encodes CFIm25, a component of mammalian cleavage factor I which regulates polyadenylation. The authors show in a variety of standard learning and memory tasks (auditory fear conditioning and performance in the water maze) that the *NUDT21*^+/-^ mice perform significantly less well than the WT mice, to different degrees, depending on the task. The authors also perform a basic analysis of "spiking" activity, based on EEG recordings and find an elevated "spiking" in one brain area. The authors then go on to analyze the effect of *NUDT21*_+/-_ on polyadenylation of mRNAs from human embryonic stem cells that are differentiated into excitatory neurons. The authors have previously published (Gennarino et al., *eLife* 2015) that *NUDT21* is associated with the usage of longer 3'UTRs in patient-derived lymphoblastoid cells.

1) Figure 4—figure supplement 1. Principal component analysis.

The principal component analysis is able to separate the control and the shRNA samples for both methodologies, 3'end sequencing and mass spectrometry. However, the 3'end sequencing control samples seem to have strong batch effects, as PC2 and PC1 are very similar. This casts doubts on the quality of the data and the validity of the results.

2) Figure 4D and Figure 4Cv.

A substantial fraction of genes (233) show an increase of mRNA length upon *NUDT21* KD. The example of *KIF9* shows an internal APA peak in the second CDS-exon under the control condition. How many of the 233 (or more) genes show the same characteristics? The authors should provide more information about their detected 3'UTR isoforms. How many of the detected peaks in the control condition correspond to known 3'UTR isoforms? What is the relative distance of all detected APA events to the known stop codon? Is there a established APA signal in the proximal upstream region of the identified peaks? Furthermore, the isoforms and their change upon *NUDT21* KD should be additionally validated by northern blot or 3'RACE for both scenarios, shortening and lengthening.

3) Figure 4E.

This figure compares the change in protein abundance (shRNA vs. control) with the change in mRNA length (shRNA vs. control). However, there is no measure of significance for the protein foldchange. The authors should show the LFQ fold changes and highlight whether the expression differences are significant or not. Furthermore, the increase/decrease in protein abundance could also be the result of changes in RNA abundance or changes in translation. This should also be addressed. How many of the 87 genes in quadrant one can be explained by the gain of a stop codon that prevents the degradation by non-stop decay?

[Editors’ note: further revisions were suggested prior to acceptance, as described below.]

Thank you for resubmitting your work entitled "Partial loss of CFIm25 causes learning deficits and aberrant neuronal alternative polyadenylation" for further consideration by *eLife*. Your revised article has been evaluated by Timothy Behrens (Senior Editor) and Sacha Nelson (Reviewing Editor) and two peer reviewers.

The manuscript has been improved but there are some remaining issues that need to be addressed before acceptance, as outlined below:

After review and further discussion, the reviewers and editors agree that the remaining two points raised by reviewer #2 need to be addressed. In the event that the statistical test suggested in point 2 is negative, the conclusions linking 3'UTR shortening to protein up-regulation should be softened or qualified.

Reviewer #1:

The authors have addressed all my concerns. I have no more comments.

Reviewer #2:

We still have a few remaining concerns:

1) Figure 4 and 6. T-SNE analysis.

We disagree with the use of t-SNE over PCA as a quality control. Both PCA and t-SNE are dimensionality reduction techniques. PCA aims to reduce dimensions, but preserves sample variance information. This is also the idea of the control plot, to visualize the variance between sequencing samples, where distances on the PCA components will correspond to differences in sample variance.

t-SNE uses a probabilistic approach to maximize sample distances to define clear sample clusters, during this process it will distort the sample variance information due to its non-linear characteristics. In the context of a control plot to visualize sample variance, t-SNE is not the right dimensionality reduction technique to use. It will artificially create better sample clusters.

2) Figure 4E and 6D. Protein analysis.

We disagree. The authors should provide a statistical analysis to test whether candidates in quadrant 2 of Figure 6D are significantly overrepresented. As is, no meaningful conclusions can be drawn about a possible relation between 3'UTR shortening and protein upregulation. Also, there are some data points (shown plotted in red) missing from their plot.

---

## [Author Response]

Essential revisions:The reviewers and editors were enthusiastic about the importance of the study, but several key concerns about the ability of the data to support the conclusions were raised. These centered mainly on the issue of sequencing and proteomics data quality and analysis. The concerns were mainly that some of the Poly(A) ClickSeq reads may be erroneous and need to be removed. False positive peaks can lead to miscalculation of genes with non-stop decay.

We took multiple steps to reduce the risk of false positive peaks and strengthen readers’ confidence in our PAC-seq results.

1) Most importantly, we resequenced all of our samples. To improve mapping, our new read length is 150 nucleotides. Further, we increased our sequencing depth in the human neurons to >45 million reads/sample, and in the mice, we reached >35 million reads/sample. This depth approaches saturation, and thus reduces variability, and importantly, provides sufficient read counts even after filtering a high percentage of internally primed reads.

2) Additionally, to reduce the risk of including false positive cleavage sites, we restricted our analysis to peaks that aligned with annotated human cleavage sites from the PolyA_DB3 database (Wang et al., 2017). This database was generated from >20 human cell types using sequencing data from libraries prepared with locked nucleic acid primers that preserved some nucleotides of the poly(A) tail in the amplicon. This allows easy identification of true cleavage sites by the presence of non-genomic adenines in the sequencing data. While we do lose some novel cleavage sites with this approach, such as with *NUDT21*, we have decided that the improved specificity to remove doubt among readers is worth it.

3) Also, to further increase the specificity of our results, we reduced our adjusted p-value cutoff from 0.1 to 0.05 in the analysis (Figure 4C and E and Figure 6C and D).

4) Lastly, we validated alternative polyadenylation changes in a few relevant genes by RT-qPCR by showing that there was relatively less of the long mRNA isoform compared to total mRNA for that gene in the *NUDT21* knockdown cells or heterozygous mice (Figure 4A and Figure 6B).

In addition to taking steps to reduce false positive peaks in the PAC-seq data, we have decided to remove the non-stop decay discussion from the manuscript. While interesting, we believe it proved a distraction from our primary point that *NUDT21* loss of function causes learning deficits and the effect is mediated by widespread misregulated APA, including in a number of genes associated with intellectual disability. Notably, though, with our new sequencing and more stringent analysis, multitudes of the identified APA genes undergo changes in non-stop decay rates. Additionally, the non-stop decay transcripts for our example genes in the initial submission, KIF9 and ZCCHC6, are annotated in the PolyA_DB3 database, and show the same lengthening and shortening. In our new source data files, we include information about where the polyadenylation sites are located in the gene (Figure 4—source data 1 and Figure 6—source data 1).

Both reviewers noted the large variation among control samples shown in Figure 4 supplement 1. The PCA plot showed variability among controls (PC2) that was nearly as large as the experimental effect.

Our new sequencing and analysis approach has resulted in less variability between the samples. Additionally, we have switched to using a t-distributed Stochastic Neighbor Embedding (t-SNE) analysis rather than the PCA. This allows for capturing multiple dimensions beyond the two visualized in a PCA and clearly reveals a separation of the genotypes in spite of some variability (Figure 4B and D, Figure 6—figure supplement 1B, and Figure 7A and D).

There were also concerns about the statistical validity of the assessment of protein abundance.

We modelled our initial quantitative mass spectrometry analysis on the method used by Brumbaugh at al. from the first example of mass spectrometry in *Nudt21* inhibited cells (Brumbaugh et al., 2018). They only used effect size, not a p-value cut off, in their analysis. This approach is common in mass spectrometry analysis, perhaps because the inherent variability of quantitative mass spectrometry is such that using p-values with multiple comparisons corrections as a cutoff will result in far too many false negatives. Moreover, that our mass spectrometry was concordant with our PAC-seq data and previous studies, provided some evidence that it is valid. However, to address this concern, we have analyzed the data and included p-values in the source data sheet (Figure 4—source data 1, Figure 6—source data 1, and Figure 7—source data 1). This analysis showed that 109 proteins from our APA analysis have a fold change with a p-adusted <0.05. However, in the scatter plot in Figure 6, we wanted to present the fold change of all of the proteins from genes with altered alternative polyadenylation to provide a clearer picture of the global effects of altered APA on protein levels (Figure 6D). We did the same for the differentially expressed genes and protein level scatter plot as well (Figure 7C).

One reviewer suggested repeating the RNA-seq for expression analysis.

PAC-seq can identify differentially expressed genes comparable to traditional RNA-seq (Elrod et al., 2019). Initially, we did not think that analysis was necessary because DEGs are likely downstream of APA changes. However, we now include those data in the manuscript along with an explanation of how DEGs are a secondary effect in the Results section (Figure 7A-C).

The other suggested performing northern blots to confirm identified 3'UTR isoforms and their differential regulation in the mutant animals.

We used RT-qPCR to confirm relative mRNA shortening of our example genes. We used primers that detected all of the mRNA for the gene along with additional primers that could only detect the long mRNA isoform. We then normalized the long isoform fold change to the total mRNA and compared the genotypes. The samples with reduced *NUDT21* showed relatively less of the long isoform, i.e. shortening (Figure 4A and Figure 6B).

They also suggested the following quality controls and metrics:- What algorithm was used to identify peaks?

From our Materials and methods section, “We computed strand-specific coverage of the features (mRNA cleavage sites) from BAM files using bedtools v2.25.0 “*genomecov”* module (Quinlan and Hall, 2010).” We essentially compute peaks as the BEDGRAPH (-bg) feature coverage summaries from the sequence alignment files (bam) for a given genome.Additional details are described in the Materials and methods section.

- What measures were taken (filtering steps) to prevent internal priming (false polyA peaks)?

In our initial submission, for potential novel p(A) sites, we scanned a 20 base-pair sliding window for genomic adenines, and excluded peaks with >12 adenines within 10 nucleotides upstream and 50 nucleotides downstream of the potential cleavage site. For our revision, to remove any doubts, we have restricted our analysis to previously annotated polyA sites from the PolyA_DB3 database, for which the authors took significant steps to prevent false positives from internal priming (described above).

- How many reads/peaks per gene feature (5'UTR/CDS/3'UTR/introns) were detected?

Over 90% of our filtered reads were in the 3′ UTR (Figure 6—figure supplement 1A). For unfiltered reads ~50% were in the 3′ UTR (see Author response image 1).

**Author response image 1. respfig1:** About 50% of unfiltered PAC-seq reads are in the 3′ UTR.

- What is the number of peaks per known poly(A) signal (how many peaks do not have known poly(A) signals)?

In our new analysis, we only included peaks mapping to a known poly(A) site in the PolyA_DB3 database. That database lists the PAS signal associated with each of its poly(A) sites (Wang et al., 2017). Out of our 8156 polyadenylation sites, 8156 (91.14%) have a poly(A) signal and 792 (0.088%) do not.

- What is the peak distance from known poly(A) signals?

From the dataset we used for our poly(A) site annotations, the peak distance from known poly(A) signals is within 40 nucleotides upstream of the cleavage site (Wang et al., 2017).

- What is the replica correlation between poly(A) peak counts?

Correlation between shScramble samples:

**Author response image 2. respfig2:** Control PAC-seq peak counts are closely correlated.

Correlation between sh*NUDT21* samples:

**Author response image 3. respfig3:** *NUDT21*knockdown PAC-seq peak counts are closely correlated.

- Are the detected peaks conserved in other species?

In the PolyA_DB3 database, Wang et al. interpret conserved poly-adenylation sites (PAS) as those that are within 24 nucleotides of one another after whole genome alignment between species. In our human neurons, 82% (36,145/44,007) of the peaks were conserved in other species

- What is the fraction of sequenced reads that fall into poly(A) peaks after filtering (It would be an issue if more reads are filtered, than reads contributing to poly(A) peaks)

The majority of reads are filtered:

SampleFractionWT10.12WT20.15WT30.10KD10.14KD20.14KD30.14

However, we don’t think this is an issue. We sequenced with sufficient depth to saturate both the real and internally primed sites, so that when we filter out the internally primed sites, we have sufficient read counts to meaningfully interpret APA. After filtering, at worst we have 4.5 million reads for a sample. Since our reads are not distributed throughout the mRNA, but rather only marking 3*'* ends, that’s more than enough.

Reviewer #1:[…]1) The authors claim that 15% of genes with altered APA may undergo non-stop decay. This is quite a substantial number. Can they rule out the possibility that this is due to technical artefact of the sequencing method?

As described above, we reduced the risk of technical artefact, we resequenced at greater depth and with longer readers. Moreover, we restricted our analysis to previously annotated cleavage sites. Even so, in the new analysis, out of 464 genes with significant altered alternative polyadenylation (P-adjusted <0.05), 113 have an internal cleavage site that has a significantly different read count in knockdown vs. controls. However, this does not necessarily mean that the internal cleavage site is the primary source or sink for shortened or lengthened mRNAs, only that it’s different between the treatment groups. Regardless, we have removed the discussion about non-stop decay because it seemed too distracting and not important for the primary purpose of the paper.

Can they verify these non-stop decay-causing poly(A) sites, either by 3'RACE (for some) or comparing their data to those from other sequencing methods?

3*'* RACE seems susceptible to the same mispriming issues as 3*'* end sequencing since it relies on a poly(T) primer. Therefore, we verified APA in example genes by RT-qPCR with primers that distinguish the long isoforms from total mRNA.

2) They also observed a non-stop decay isoform for NUDT21. However, this does not appear congruent with the fact that there is 30% reduction of protein level but 50% mRNA reduction. Non-stop decay should reduce both mRNA and protein expression levels. Therefore, the mRNA reduction should be more than 50%. Also, how can they explain the increase protein production per mRNA?

The increased protein relative to mRNA expression in *NUDT21* is certainly enigmatic. Our initial hypothesis that *NUDT21* knockdown caused a relative increase in *NUDT21* mRNAs with shorter 3*'* UTRs that are translated more efficiently turned out to be incorrect, as confirmed by the PAC-seq data. Our current leading hypothesis is that the CFIm25 protein is degraded more slowly when its levels are lower, perhaps because it’s more likely to be bound in its complex. The presence of non-stop decay transcripts would not affect this mechanism. The experiments to test this, while interesting, do not help answer the questions of this manuscript: can *NUDT21* loss of function alone cause learning deficits and what molecular pathology may mediate the symptoms? Thus, we did not pursue these experiments. Moreover, we did not include the non-stop transcripts for *NUDT21* in the new analysis because they were not previously annotated.

Reviewer #2:[…]1) Figure 4—figure supplement 1. Principal component analysis.The principal component analysis is able to separate the control and the shRNA samples for both methodologies, 3'end sequencing and mass spectrometry. However, the 3'end sequencing control samples seem to have strong batch effects, as PC2 and PC1 are very similar. This casts doubts on the quality of the data and the validity of the results.

We sequenced all (the independent) samples in the same batch for both the PAC-seq and mass spectrometry experiments, so there can be no batch effects affecting just some of the samples. To reduce variability, we resequenced our samples at greater depth and analyzed the global data with t-SNE so as to capture all the dimensions of the data in the plot. The genotypes segregate in all of the experiments (Figure 4B and D, Figure 6—figure supplement 1B, and Figure 7A and D).

2) Figure 4D + Figure 4Cv.A substantial fraction of genes (233) show an increase of mRNA length upon NUDT21 KD. The example of KIF9 shows an internal APA peak in the second CDS-exon under the control condition. How many of the 233 (or more) genes show the same characteristics?

Of the 89 genes that show lengthening in our new, more stringent, analysis, 37 have reads that map to previously annotated internal poly(A) sites (Figure 6—source data 1). In the source data spreadsheet, we now indicate if there is an internal poly(A) site and we list the different regions with poly(A) sites for each gene, e.g. 5*'* UTR, CDS, intron, and 3*'* UTR. However, because a gene has non-stop decay transcripts, it doesn’t require that those transcript levels are changing between the test groups. Some do, but for others, the change could be between different poly(A) sites within the 3*'* UTR. For *KIF9* in the new data set, the wild type neurons only have the internal peak with non-stop decay transcripts, whereas the sh*NUDT21*-infected neurons have peaks in both the intron and the 3*'* UTR, indicating a lengthening as we saw in the initial submission.

Relatedly, ~3% of our filtered reads mapped to a CDS-exon and ~2% to introns (Figure 6—figure supplement 1A).

The authors should provide more information about their detected 3'UTR isoforms. How many of the detected peaks in the control condition correspond to known 3'UTR isoforms?

With our current analysis, we used previously annotated sites as a filter, thus 100% of the peaks correspond to known isoforms.

What is the relative distance of all detected APA events to the known stop codon?

We have removed the discussion about non-stop decay.

Is there an established APA signal in the proximal upstream region of the identified peaks?

The polyA_DB3 database which we used for our peak annotations lists proximal upstream poly(A) signals for all of its poly(A) sites (Wang et al., 2017). Most sites have one of the canonical hexamers, but some have no poly(A) signal.

Furthermore, the isoforms and their change upon NUDT21 KD should be additionally validated by northern blot or 3'RACE for both scenarios, shortening and lengthening.

3*'* RACE is vulnerable to the same mispriming as 3*'* end sequencing and the expression levels of our more controversial calls, such as *KIF9*, are probably too low to be detected by Northern blot. Therefore, as described above, to confirm the validity of poly(A) site calls, we resequenced our samples with long reads and greater depth and only included peaks that mapped to previously annotated cleavage sites in our analysis. Further, we confirmed the levels of the different isoforms of example genes by RT-qPCR. We chose our example genes for their disease relevance and history in *NUDT21* research.

3) Figure 4E.This figure compares the change in protein abundance (shRNA vs. control) with the change in mRNA length (shRNA vs. control). However, there is no measure of significance for the protein foldchange. The authors should show the LFQ fold changes and highlight whether the expression differences are significant or not.

We include P-values and adjusted P-values (Figure 6—source data 1). 109 proteins have an adjusted p-value <0.05. However, in the chart we include all the proteins from genes with altered APA because we are most interested in general trends, i.e. the majority of affected genes shorten and have a corresponding increase in protein levels. Because of the inherent variability in quantitative mass spectrometry and to correct for multiple comparisons in so many proteins, we believe using an adjusted p-value cutoff causes too many false negatives and vitiates our overall interpretation of the data.

Furthermore, the increase/decrease in protein abundance could also be the result of changes in RNA abundance or changes in translation. This should also be addressed.

Our initial purpose in that figure was to show what happened to the protein abundance of the genes with misregulated APA. However, we’re now including differentially expressed gene data and how that correlates with protein levels (Figure 7A-C).

How many of the 87 genes in quadrant one can be explained by the gain of a stop codon that prevents the degradation by non-stop decay?

Of the 89 genes that show lengthening in our new, more stringent, analysis, 39 have reads that map to previously annotated internal poly(A) sites that have significantly lower read counts in the sh*NUDT21*-infected cells (Figure 6—source data 1). This does not necessarily mean that all of the lengthening is coming from non-stop decay transcripts, though, only some of them. As stated above, In the source data spreadsheet, we now indicate if there is an internal poly(A) site and we list the different regions with poly(A) sites for each gene, e.g. 5*'* UTR, CDS, intron, and 3*'* UTR.

[Editors’ note: what follows is the authors’ response to the second round of review.]

Reviewer #2:We still have a few remaining concerns:1) Figure 4 and 6. T-SNE analysis.We disagree with the use of t-SNE over PCA as a quality control. Both PCA and t-SNE are dimensionality reduction techniques. PCA aims to reduce dimensions, but preserves sample variance information. This is also the idea of the control plot, to visualize the variance between sequencing samples, where distances on the PCA components will correspond to differences in sample variance.t-SNE uses a probabilistic approach to maximize sample distances to define clear sample clusters, during this process it will distort the sample variance information due to its non-linear characteristics. In the context of a control plot to visualize sample variance, t-SNE is not the right dimensionality reduction technique to use. It will artificially create better sample clusters.

Our purpose for presenting the t-SNE charts was simply to show that the data separated by genotype. PCA accomplishes this as well, and readers will be more familiar with it, so we have switched all of the t-SNE charts to PCA plots as the reviewer requested and have changed the text accordingly (Figures 4, 6, and 7).

2) Figure 4E and 6D. Protein analysis.We disagree. The authors should provide a statistical analysis to test whether candidates in quadrant 2 of Figure 6D are significantly overrepresented. As is, no meaningful conclusions can be drawn about a possible relation between 3'UTR shortening and protein upregulation.

For Figure 6, to calculate a p-value for enrichment of mRNA shortening after partial NUDT21 loss in Figure 6C, we used a two-tailed, chi-square test (P< 0.0001). To calculate a p-value for the enrichment of shorter mRNAs and increased protein in Figure 6D, we first performed a two-tailed, chi-square test with all the data (quadrants I-IV) to show that they were not evenly distributed (P<0.0001), then performed a conditional chi-square test with only the protein levels from genes with shorter mRNAs (quadrants II and III) to show that mRNA shortening more often results in elevated protein levels (P< 0.0001). Together, these tests show that *NUDT21* loss predominantly results in shorter mRNAs and elevated protein levels. The p-values can be found in the legend for Figure 6, and the explanation of the conditional chi-square analysis is in the mass spectrometry analysis section of the methods.

For Figure 4E in the text, we changed our description to, “Although there are too few data points to perform a statistical analysis, all observations follow the trend seen in cell lines, where protein levels increase with mRNA shortening.”

Also, there are some data points (shown plotted in red) missing from their plot.

We admire the thoroughness of the reviewer and greatly appreciate them catching those outlier data points. Those data were false positives and should have been excluded. Almost all of them only had a single detected peptide in one sample from each genotype. In the majority of samples from both genotypes, those proteins were undetected. When that close to the threshold of detection, fold change calculations misrepresent the truth: the fold change can be computed as enormous for an insignificant or incidental difference in protein level. Thus, we have removed proteins from the analysis that did not have reproducible expression in at least one genotype. Moreover, so that proteins with undetected peptide in some samples did not get artificially amplified fold change calculations, we used an imputation strategy from Perseus, a popular proteomics analysis suite (Tyanova et al., 2016). We imputed missing values with random draws from a Gaussian distribution downshifted two standard deviations from the mean of all the measured values, keeping the scaling factor constant at one. This serves to preserve the empirically measured variance while selecting reasonable values near the limit of detection, and avoids artificially underestimating variance in proteins that are completely undetected in one genotype. Our updated analysis has fewer false positives and strikes a better balance between specificity and sensitivity.